# Changes in the amount of nutrient of packaged foods and beverages after the initial implementation of the Chilean Law of Food Labelling and Advertising: A nonexperimental prospective study

**Marcela Reyes**[1], **Lindsey Smith Taillie**[2], **Barry Popkin**[2], **Rebecca Kanter**[3], **Stefanie Vandevijvere**[4,5], **Camila Corvalán**[1] *

**1** Institute of Nutrition and Food Technology (INTA), University of Chile, Santiago, Chile, **2** Department of Nutrition and Carolina Population Center, University of North Carolina at Chapel Hill, Chapel Hill, North Carolina, United States of America, **3** Department of Nutrition, Faculty of Medicine, University of Chile, Santiago, Chile, **4** Department of Epidemiology and Biostatistics, School of Population Health, University of Auckland, Auckland, New Zealand, **5** Scientific Institute of Public Health (Sciensano), Department of Epidemiology and Public Health, Brussels, Belgium

* ccorvalan@inta.uchile.cl

**Data Availability Statement:** The datasets used and/or analyzed during the current study cannot be

## Abstract

### Background

In June 2016, the first phase of the Chilean Food Labelling and Advertising Law that mandated front-of-package warning labels and marketing restrictions for unhealthy foods and beverages was implemented. We assess foods and beverages reformulation after this initial implementation.

### Methods and findings

A data set with the 2015 to 2017 nutritional information was developed collecting the information at 2 time periods: preimplementation (T0: January–February 2015 or 2016; $n = 4,055$) and postimplementation (T1: January–February 2017; $n = 3,025$). Quartiles of energy and nutrients of concern (total sugars, saturated fats, and sodium, per 100 g/100 mL) and the proportion of products with energy and nutrients exceeding the cutoffs of the law (i.e., products "high in") were compared pre- and postimplementation of the law in cross-sectional samples of products with sales >1% of their specific food or beverage groups, according to the Euromonitor International Database; a longitudinal subsample (i.e., products collected in both the pre- and postimplementation periods, $n = 1,915$) was also analyzed. Chi-squared, McNemar tests, and quantile regressions (simple and multilevel) were used for comparing T0 and T1. Cross-sectional analysis showed a significant decrease (T0 versus T1) in the proportion of product with any "high in" (from 51% [95% confidence interval (CI) 49–52] to 44% [95% CI 42–45]), mostly in food and beverage groups in which regulatory cutoffs were below the 75th percentile of the nutrient or energy distribution. Most frequent reductions were in the proportion of "high in" sugars products (in beverages, milks and milk-

made available. The 2015-2017 nutritional information dataset was obtained upon a legal agreement made with the supermarket association ASACH. Such agreement includes a clause of not making publicly available the data. Contact: https://www.supermercadosdechile.cl/contacto/. The Euromonitor International Database set is a commercial database that can be obtained upon payment from https://www.euromonitor.com/.

**Funding:** Funding for this study was obtained from Bloomberg Philanthropies (PI: BP), IDRC (#107731-002 PI: CC), and Fondecyt (#3150183; PI: RK). Authors have not received payment to write this article by a pharmaceutical company or other agency. The corresponding author had full access to all the data in the study and all authors shared final responsibility for the decision to submit for publication. IDRC URL: https://www.idrc.ca/ Bloomberg: https://www.bloomberg.org/ CONICYT: https://www.conicyt.cl/ The funders had no role in study design, data collection and analysis, decision to publish, or preparation of the manuscript.

**Competing interests:** I have read the journal's policy and the authors of this manuscript have the following competing interests: BP is an Academic Editor for PLOS Medicine. We have no other competing interest to disclose.

**Abbreviations:** FOP, front of package; NCD, noncommunicable disease.

based drinks, breakfast cereals, sweet baked products, and sweet and savory spreads; from 80% [95% CI 73–86] to 60% [95% CI 51–69]) and in "high in sodium" products (in savory spreads, cheeses, ready-to-eat meals, soups, and sausages; from 74% [95% CI 69–78] to 27% [95% CI 20–35]). Conversely, the proportion of products "high in" saturated fats only decreased in savory spreads ($p < 0.01$), and the proportion of "high in" energy products significantly decreased among breakfast cereals and savory spreads (both $p < 0.01$). Quantile analyses showed that most of the changes took place close to the cutoff values, with only few exceptions of overall left shifts in distribution. Longitudinal analyses showed similar results. However, it is important to note that the nonexperimental nature of this study does not allow to imply causality of these findings.

## Conclusions

Our results show that, after initial implementation of the Chilean Law of Food Labelling and Advertising, there was a significant decrease in the amount of sugars and sodium in several groups of packaged foods and beverages. Further studies should clarify how food reformulation will impact dietary quality of the population.

## Author summary

### Why was this study done?

- Reformulation of processed foods and beverages has been defined as one of the most cost-effective measures for preventing obesity and cardiometabolic diseases.

- In June 2016, Chile implemented the first phase of Food Labelling and Advertising Law that mandates the use of front-of-package warning labels, marketing restrictions for unhealthy foods and beverages, and banning of sales at school.

- A previous study showed no relevant reformulation in anticipation to the initial implementation of the law, but Chilean food and beverage companies had claimed they reformulated 20% of packaged foods after implementation.

### What did the researchers do and find?

- We collected information of the nutrient fact panels of packaged foods and beverages previous to the implementation of the law (2015 and 2016; 4,055 items) and <1 year after initial implementation (2017; 3,025 items).

- We found a significant decrease in the proportion of foods and beverages considered as unhealthy ("high in" energy, sugars, saturated fats, or sodium) from 51% to 44%, mostly in food and beverage groups in which regulatory values were below the 75th percentile of the nutrient or energy distribution.

- Most frequent reductions were in the proportion of "high-in" sugars products (beverages, milks and milk-based products, breakfast cereals, sweet baked products, and sweet and savory spreads) and "high in" sodium products (savory spreads, cheeses, ready-to-eat meals, sausages, and soups) whereas "high in" saturated fat reductions only took

place in savory spreads and "high in" energy among breakfast cereals and savory spreads.

- Quantile regression analyses showed that most of the changes occurred around regulation cutoff values, with minor shifts on overall energy or nutrient distributions.

### What do these findings mean?

- After initial implementation of the first phase of the Chilean Food Labelling and Advertising Law, we observed important decreases in the amount of sugars and sodium in several groups of packaged foods and beverages.

- Future follow-ups should address the sustainability of such improvements and whether the reported changes translate into healthier diets.

## Introduction

From 1990–2020, packaged foods and beverages with a high degree of processing have become increasingly available worldwide [1, 2]. Those foods and beverages usually have a high amount of energy and nutrients that have been linked to a higher risk of noncommunicable diseases (NCDs; i.e., saturated fats, sugars, and sodium, here forward referred to as nutrients of concern), which contribute to the global burden of disease associated with poor diets [1, 3].

Improving the nutritional quality of packaged foods and beverages, specifically by decreasing the amount of nutrients of concern, has been a major issue in nutrition policy since the second half of the 20th century [4, 5]. Recent reports suggest that reformulation is the most cost-effective measure to improve populations' diets and health status [4, 6, 7]; although some authors question whether reformulation will result in a significant improvement of the overall nutritional quality of the diet [8]. Different voluntary or regulatory strategies may incentivize reformulation: setting standards for nutrient amount (e.g., banning *trans* fats or adding upper limits for the amount of sodium) [5, 9–16], implementing fiscal policies, or adding easy-to-understand front-of-package (FOP) nutrition labeling on packaged foods and beverages [17–21], among others. However, as most of these strategies are voluntary, industry is less likely to comply, limiting the impact of the measures [9, 10, 18, 20, 22, 23]. In June 2016, Chile implemented the Law of Food Labelling and Advertising (hereafter, the law) that mandates that packaged foods and beverages with added sugars, saturated fats, or sodium that are above the established cutoffs of nutrients of concern or energy must display up to 4 FOP warning labels that say "high in [nutrient of concern]" (hereafter, products "high in"). The law was implemented in a staggered way, with cutoffs becoming increasingly stricter over 3 phases (S1 Table). Products "high in" cannot be sold or distributed in the school food environment nor be marketed to children under 14 years of age [24]. The Chilean government has introduced a package of strategies to prevent childhood obesity that combines a FOP warning label with actions that discourage the consumption of unhealthy foods by children, which are hypothesized to show a more extended impact on food reformulation than single policies. Therefore, in the current study, we aimed to study changes in the proportion of "high in" products and changes in energy and nutrients of concern in packaged foods and beverages before (2015–2016) and after (2017) the initial implementation (i.e., <1 year of the first phase cutoffs) of the Chilean law.

## Methods

### Summary of study design

A nutritional information 2015 to 2017 data set was developed with data collected in supermarkets from Santiago (where 30% of the population in Chile lives), Chile, in periods pre- (T0: January–February 2015 or 2016; $n$ = 4,055) and postimplementation (T1: January–February 2017; $n$ = 3,025) of the law. Nutrient information declared on the food labels was compared between T0 and T1. The analytical sample included packaged foods and beverages with sales ≥1% of their specific food groups, according to Euromonitor International Database [25]. The same outcomes were studied in a longitudinal subsample (i.e., products collected in both the pre- and postimplementation periods; $n$ = 1,915). In both analyses, comparisons included the amount of energy and nutrients of concern and the proportion of products "high in" (i.e., foods and beverages with the amount of energy and nutrients above the initial cutoffs). The analyses plan was developed prospectively (S1 Text) with later adjustments according to reviewer's suggestions (i.e., quantile regressions). This study is reported as per the Strengthening the Reporting of Observational Studies in Epidemiology (STROBE) guideline (See S1 and S2 Checklists).

### Collection of nutritional information and ingredients of packaged foods and beverages

Data collection occurred during 3 waves—January to February 2015, January to February 2016, and January to February 2017—at 6 major supermarkets (1 supermarket from each of the 6 major chains in Chile; about 60% of food retail) in high-income neighborhoods in Santiago, Chile, known to have a great variety of food products. Three candy distributors were also included in order to increase the variety of candies and sweet confectioneries. Data were initially collected as part of the multinational collaborative effort for monitoring food environment, INFORMAS [26], upon an agreement with the Chilean National Association for Supermarkets (ASACH). Photos of all packaged foods and beverages available in the store where obtained (i.e., products packaged before being sold, which exclude bulk foods as well as food chosen at the counter). When multiple package sizes were available, only the largest package was collected. The entire package was photographed. After each data collection wave, trained dietitians reviewed the photos and entered general identifying information separately for each product (i.e., barcode, brand, flavor or other important identifier details, manufacturer, etc.), the ingredients list, amount of energy and nutrients (i.e., protein, carbohydrates, total sugars, total fats, fat subtypes if available, and sodium) per 100 g or 100 mL. When implausible information on the amount of energy and nutrients was detected (i.e., addition of grams of macronutrients per 100 g > 100 g or addition of grams of carbohydrates per 100 g × 4 + grams of proteins per 100 g × 4 + grams of total fats per 100 g × 9 < 0.9 or >1.1 declared energy [kcal] per 100 g), pictures were re-reviewed for corrections when possible. When applicable, the instructions for product reconstitution (e.g., powdered milks, condensed juices, etc.) were also entered because they are needed for estimating the amount of energy and nutrients of the product as consumed. In Chile, declaration of nutrients and ingredients are mandatory for packaged foods and beverages, both per serving size and per 100 g or 100 mL.

### Food groups

Foods and beverages were assigned to one of 17 mutually exclusive groups, based on a previously used classification [27]. Groups for this analysis were: beverages (sugar-sweetened, non–

sugar-sweetened, and unsweetened); milks and milk-based drinks; yogurts; breakfast cereals (ready-to-eat and to be prepared); sweet baked products; desserts and ice creams; candies and sweet confectionery; sweet spreads; savory baked products; savory snacks; savory spreads; cheeses; ready-to-eat meals; sausages; nonsausage meat products; and soups (powder and ready-to-eat). Examples of products included in every group are shown in S2 Table.

## Data processing and definition of the analytical sample

Fig 1 shows the number of products that were either included or eliminated from the analytical samples. A total of 26,748 products were photographed during the 3 data collection waves between 2015 and 2017. Data collected in years 2015 and 2016 were pooled for constructing a more comprehensive preimplementation (T0) sample; in the case of duplicate products (intra- or interyear), only the most recent product was retained (i.e., only items collected in 2016 were included). Data collected in 2017 constituted the postimplementation (T1) sample; duplicated products were also eliminated for this period. From both cross-sectional samples (T0 and T1), we excluded items that lacked relevant information (i.e., did not include the ingredients list, any information on the amount of energy and nutrients, reconstitution instructions when needed; 3.3% for T0 and 2.7% for T1), were not under the scope of the regulation (i.e., unprocessed and minimally processed foods and culinary ingredients with no increase in the natural content of nutrient of concern as part of their processing; 14.1% for T0 and 15.5% for T1), and were not included among the best-selling products (i.e., <1% market share within each of the 52 main food groups from the Euromonitor database [25]; 49.4% for T0 and 52.9% for T1). For T0 and T1, market share within a specific food group was computed as Euromonitor sales of <product or brand family of products> during <year> × 100 ÷ addition of sales of <Euromonitor food group>. Products with ≥1% of the market share of their food group were selected manually from the list of Euromonitor products (or brand family of products if product was not directly available).

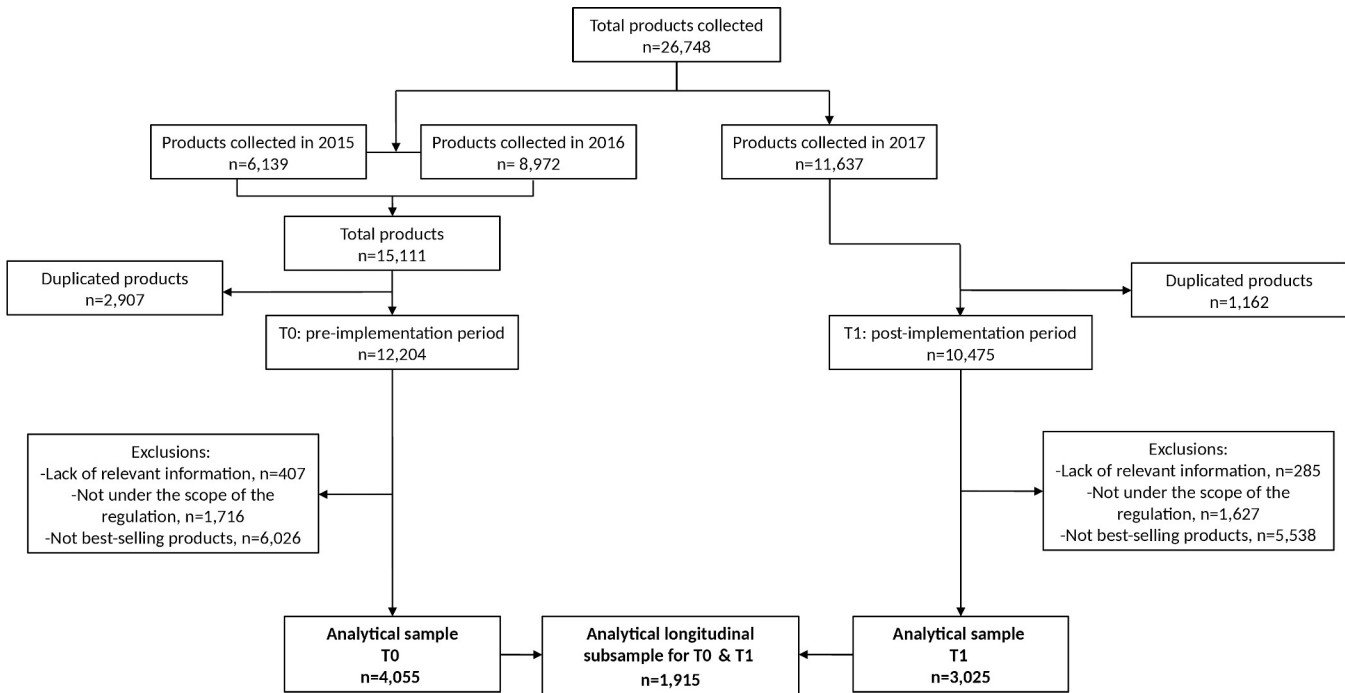

**Fig 1. Flow chart describing products excluded from the analytical sample.** T0, preimplementation period; T1, postimplementation period.

In line with the Chilean nutrient declaration labeling rules, missing values for saturated fats were replaced by 0 when the amount of total fats was below 3 g per portion size (for missing values other than saturated fats, the missing value was not assigned to any imputed value, nor was the food or beverage item eliminated from the analysis). Additionally, 154 implausible values for the amount of energy and nutrients of concern were omitted prior to the statistical analyses. Details on the exclusion criteria are presented in S3 Table.

Classification of "high in" products was done among those under the scope of the regulation (i.e., products with added sugars, saturated fats or sodium; the list of specific ingredients which could add those nutrients has been reported previously) [28]. Cutoffs for solids or liquids (S1 Table) were used depending on the unit of measure displayed on the label—g or mL, respectively. Given the heterogeneity of unit of measure, however, used among ice creams, some spreads (both sweet and savory) and yogurts, we standardized the nutrient amount to 100 g for these product types by considering the following food specific gravities: 0.5 g/mL for ice creams, 1.15 g/mL for desserts, 1.15 g/mL for mustard, 0.95 g/mL for mayonnaise, and 1.06 g/mL for yogurts and subsequently used the "high in" cutoffs for solids.

A subsample of products that were collected in both T0 and T1 were longitudinally analyzed (approximately 50% of T1 belonged to T0, $n$ = 1,915) to specifically address reformulation (i.e., these analyses exclude the effect of products entering or exiting the food market or potential sampling differences between waves). To construct both the cross-sectional and the longitudinal analytical samples, matching products between and within waves were identified programmatically based on barcode, brand, manufacturer name, and important identifier details (i.e., flavor).

## Data analyses

**Study outcomes.** The main outcomes were changes in (i) the proportion of products exceeding the cutoffs of the initial phase of the law (i.e., "high in" products) for energy, total sugars, saturated fats, sodium, or any "high in" (i.e., products "high in" energy or at least one nutrient of concern) [24], by food or beverage group, between the pre- and postimplementation periods and (ii) the quartiles of energy, total sugars, saturated fats, and sodium (amount per 100 g or 100 mL), by food or beverage group; we used quartiles because variables were not normally distributed.

**Statistical analyses.** Changes in the proportion of "high in" foods or beverages between T0 and T1 were examined differently depending on the analysis type. The chi-square test was used for the cross-sectional analysis, and the McNemar test was used for the longitudinal analysis. In the cross-sectional analyses, we used quantile regressions with implementation period as the independent variable for estimating changes in every quartile (i.e., 25th, 50th, and 75th percentiles) of energy or the nutrient of concern by food or beverage group; in the longitudinal analyses, we used quantile regression for linear mixed-effect models (fixed or random effect were selected according to the Hausman test and Akaike's information criterion). Quantile regressions for the overall sample were performed, considering food or beverage groups and the interaction of such groups with the implementation period as covariate. Because interactions were significant ($p < 0.1$), further analyses were done stratifying by food and beverage groups. Other confounders and effect modifiers were not assessed. Size of the cross-sectional analytical sample was defined by products available at each data collection wave and the inclusion and exclusion criteria. Food groups sizes ranged from 69 to 482 products; the minimum size allows detecting a 50% relative decrease in the proportion of "high in" products and a change of 55% in the standard deviation between the pre- and postimplementation period (alpha error 0.05 and beta error 0.2). All analyses were performed using STATA V15 (release

15; StataCorp LP, College Station, TX; http://www.stata.com), except for the multilevel quantile regressions which were done using R (release 4.0.0; R Foundation for Statistical Computing).

## Results

In Table 1, we observed that, in the cross-sectional analyses, the overall proportion of products with any "high in" warnings significantly decreased from 51% (95% confidence interval [CI] 49–52) to 44% (95% CI 42–45) after the initial implementation of the law. Table 1 shows changes in the proportion of "high in" products by food and beverage groups. Decreases were most common in "high in" sugars (for beverages, milks and milk-based drinks, breakfast cereals, sweet baked products, sweet and savory spreads [all $p < 0.01$]) and "high in" sodium (savory spreads, cheeses, sausages, and soups [all $p < 0.01$], and ready-to-eat meals [$p = 0.03$]). Conversely, the proportion of products "high in" saturated fats only decreased in savory spreads ($p < 0.01$), and the proportion of products "high in" energy significantly decreased among breakfast cereals and savory spreads (both $p < 0.01$). Changes took place in food and beverage groups in which the regulatory cutoffs were below the 75th percentile of the nutrient or energy distribution; except in a few cases, mostly for saturated fats cutoffs (sweet baked products, candies and sweet confectioneries, cheeses, and sausages) and for sugars cutoffs (desserts and ice creams and candies and sweet confectioneries).

Figs 2–17 show distributions of energy and nutrients of concern by food or beverage group for T0 and T1 (cross-sectional samples). Food and beverage groups with a significant decrease in the proportion of "high in" products showed a left shift of the T1 distribution (blue curve) below the initial cutoffs (red line). In Table 2, we present the quantile regression analyses by food and beverage groups. We show that, for all food groups in which there were significant decreases in "high-in" sugars, there were also significant decrease in the quantile closer to the regulation cutoffs ($p < 0.01$ for the 75th percentile in beverages, milk and milk-based drinks, and breakfast cereals; $p = 0.03$ for median in sweet spreads) but not necessarily in the other quantiles. The exception was given by sweet baked products ($p = 0.05$ for 25th percentile) and savory spreads ($p = 0.77$ for 75th percentile). Similarly, food groups in which "high in" sodium significantly decreased, showed a significant decrease in the quantile closer to the cutoff: $p < 0.01$ for 75th percentile in savory spreads and 25th percentile in sausages; although, there was no significant change in the case of cheeses ($p = 0.11$ for 75th percentile), ready-to-eat meals ($p = 0.21$ for 75th percentile), and soups ($p = 0.18$ for 25th percentile). Same happened for energy in the case of breakfast cereals ($p < 0.01$ for the 25th percentile decrease), and savory spreads ($p < 0.01$ for the 50th percentile), as well as in savory spreads (with a decrease in the 50th percentile of saturated fats, $p = 0.02$, respectively). On the other hand, some food and beverages groups presented significant right shifts in distribution of sugars (sausages and nonsausage meat products), sodium (breakfast cereals and desserts and ice creams), saturated fats (yogurts, desserts and ice creams, sweet spreads, ready-to-eat meals, and nonsausage meat products), and energy (savory spreads).

In Table 3, we observe that in the longitudinal subsample there was also a significant decrease in the proportion of any "high in" from 52% (95% CI 49–54) to 42% (95% CI 40–44; $p < 0.01$). Food groups in which we observed significant decreases in "high in" sugars, "high in" sodium, "high in" saturated fats, and "high in" energy were all the same than in the cross-sectional analyses, except for sweet baked products and savory spreads in the case of sugars, and ready-to-eat meals and soups, in the case of sodium.

Figs 18–33 show distributions of energy and nutrients of concern by food or beverage group for T0 and T1 for the longitudinal subsample. Figures show a left shift of the T1

**Table 1. Changes between T0 and T1 in the proportion of "high in" energy and nutrients of concern (or any "high in") by food and beverage group, cross-sectional analysis.**

| Food or beverage | T0, % (95% CI) | T1, % (95% CI) | p-value | Relative change, % of T0 |
|---|---|---|---|---|
| **Beverages** | **n = 686** | **n = 482** | | |
| Any "high in" | **26 (23–29)** | **11 (8–14)** | **<0.01** | −58 |
| High in energy (T0 cutoff: 99th percentile) | 0.1 (0.004–0.8) | 0 (0–1) | 0.40 | −100 |
| High in sugars (T0 cutoff: 64th percentile) | **26 (23–29)** | **11 (8–14)** | **<0.01** | −58 |
| High in saturated fats (T0 cutoff: NA) | 0 | 0 | NA | NA |
| High in sodium (T0 cutoff: 99th percentile) | 0 | 0 | NA | NA |
| **Milks and milk-based drinks** | **n = 201** | **n = 103** | | |
| Any "high in" | **32 (26–39)** | **2 (0.2–7)** | **<0.01** | −94 |
| High in energy (T0 cutoff: 99th percentile) | 0.5 (0.01–2.7) | 0 | 0.47 | −100 |
| High in sugars (T0 cutoff: 56th percentile) | **32 (25–39)** | **2 (0.2–7)** | **<0.01** | −94 |
| High in saturated fats (T0 cutoff: 99th percentile) | 0 | 0 | NA | NA |
| High in sodium (T0 cutoff: 96th percentile) | 0.5 (0.01–3) | 0 | 0.47 | −100 |
| **Yogurts** | **n = 312** | **n = 272** | | |
| Any "high in" | 0 | 0 | NA | NA |
| High in energy (T0 cutoff: 99th percentile) | 0 | 0 | NA | NA |
| High in sugars (T0 cutoff: 99th percentile) | 0 | 0 | NA | NA |
| High in saturated fats (T0 cutoff: 99th percentile) | 0 | 0 | NA | NA |
| High in sodium (T0 cutoff: 99th percentile) | 0 | 0 | NA | NA |
| **Breakfast cereals** | **n = 148** | **n = 125** | | |
| Any "high in" | **80 (73–86)** | **61 (52–69)** | **<0.01** | −25 |
| High in energy (T0 cutoff: 14th percentile) | **81 (74–87)** | **61 (52–69)** | **<0.01** | −25 |
| High in sugars (T0 cutoff: 53rd percentile) | **46 (38–55)** | **24 (17–33)** | **<0.01** | −48 |
| High in saturated fats (T0 cutoff: 89th percentile) | 8 (4–14) | 7 (3–13) | 0.65 | −13 |
| High in sodium (T0 cutoff: 99th percentile) | 0 | 0.8 (0.2–4) | 0.27 | NA |
| **Sweet baked products** | **n = 198** | **n = 173** | | |
| Any "high in" | **100** | **94 (89–97)** | **<0.01** | −6 |
| High in energy (T0 cutoff: 6th percentile) | 93 (89–96) | 91 (86–95) | 0.44 | −2 |
| High in sugars (T0 cutoff: 6th percentile) | **95 (90–97)** | **83 (77–89)** | **<0.01** | −13 |
| High in saturated fats (T0 cutoff: 23rd percentile) | 75 (68–81) | 66 (59–73) | 0.07 | −12 |
| High in sodium (T0 cutoff: 99th percentile) | 0 | 0 | NA | NA |
| **Desserts and ice creams** | **n = 437** | **n = 333** | | |
| Any "high in" | 43 (38–48) | 42 (37–48) | 0.90 | −2 |
| High in energy (T0 cutoff: 96th percentile) | 3 (2–5) | 3 (1–5) | 0.82 | 0 |
| High in sugars (T0 cutoff: 65th percentile) | 35 (30–40) | 33 (28–38) | 0.60 | −6 |
| High in saturated fats (T0 cutoff: 75th percentile) | 24 (20–28) | 26 (21–31) | 0.58 | +8 |
| High in sodium (T0 cutoff: 99th percentile) | 0 | 0 | NA | NA |
| **Candies and sweet confectioneries** | **n = 391** | **n = 445** | | |
| Any "high in" | **87 (83–90)** | **82 (78–85)** | **0.04** | −6 |
| High in energy (T0 cutoff: 27th percentile) | 73 (68–77) | 70 (66–74) | 0.40 | −4 |
| High in sugars (T0 cutoff: 19th percentile) | 80 (76–84) | 76 (71–80) | 0.11 | −5 |
| High in saturated fats (T0 cutoff: 47th percentile) | 52 (47–57) | 47 (42–51) | 0.14 | −10 |
| High in sodium (T0 cutoff: 99th percentile) | 0.3 (0.006–1.4) | 0.2 (0.006–1.3) | 0.93 | −33 |
| **Sweet spreads** | **n = 165** | **n = 115** | | |
| Any "high in" | **75 (67–81)** | **62 (52–71)** | **0.02** | −17 |
| High in energy (T0 cutoff: 99th percentile) | 1.8 (0.4–5) | 5.2 (1.9–11) | 0.12 | +189 |
| High in sugars (T0 cutoff: 37th percentile) | **60 (53–68)** | **41 (32–51)** | **<0.01** | −32 |

*(Continued)*

**Table 1.** (Continued)

| Food or beverage | T0, % (95% CI) | T1, % (95% CI) | p-value | Relative change, % of T0 |
|---|---|---|---|---|
| High in saturated fats (T0 cutoff: 83rd percentile) | 17 (12–24) | 26 (18–35) | 0.08 | +53 |
| High in Sodium (T0 cutoff: 99th percentile) | 0 | 0 | NA | NA |
| **Savory baked products** | **n = 100** | **n = 81** | | |
| Any "high in" | 64 (54–73) | 65 (54–76) | 0.84 | +2 |
| High in energy (T0 cutoff: 32rd percentile) | 64 (53–73) | 65 (54–76) | 0.80 | +2 |
| High in sugars (T0 cutoff: 94th percentile) | 5 (2–11) | 4 (1–11) | 0.69 | −20 |
| High in saturated fats (T0 cutoff: 86th percentile) | 11 (7–19) | 7 (3–15) | 0.41 | −36 |
| High in sodium (T0 cutoff: 93rd percentile) | 6 (2–13) | 4 (1–11) | 0.45 | −33 |
| **Savory snacks** | **n = 69** | **n = 70** | | |
| Any "high in" | **94 (86–98)** | **100** | **0.04** | +6 |
| High in energy (T0 cutoff: 3rd percentile) | 91 (82–97) | 99 (92–100) | 0.05 | +9 |
| High in sugars (T0 cutoff: 99th percentile) | 0 | 0 | NA | NA |
| High in Saturated fats (T0 cutoff: 80th percentile) | 16 (8–27) | 7 (2–16) | 0.10 | −56 |
| High in sodium (T0 cutoff: 94th percentile) | 6 (2–14) | 3 (0.03–9) | 0.39 | −50 |
| **Savory spreads** | **n = 210** | **n = 174** | | |
| Any "high in" | **75 (69–81)** | **48 (41–56)** | **<0.01** | −36 |
| High in energy (T0 cutoff: 56th percentile) | **43 (36–50)** | **26 (20–33)** | **<0.01** | −38 |
| High in sugars (T0 cutoff: 92nd percentile) | **8 (4–12)** | **2 (0.4–5)** | **<0.01** | −75 |
| High in saturated fats (T0 cutoff: 56th percentile) | **44 (37–51)** | **28 (21–35)** | **<0.01** | −33 |
| High in sodium (T0 cutoff: 65th percentile) | **33 (27–40)** | **18 (13–24)** | **<0.01** | −45 |
| **Cheeses** | **n = 109** | **n = 117** | | |
| Any "high in" | 81 (72–88) | 86 (79–92) | 0.26 | +6 |
| High in energy (T0 cutoff: 75th percentile) | 24 (16–33) | 23 (16–32) | 0.89 | −4 |
| High in sugars (T0 cutoff: 99th percentile) | 0 | 0 | NA | NA |
| High in saturated fats (T0 cutoff: 20th percentile) | 80 (71–87) | 85 (78–91) | 0.26 | 6 |
| High in sodium (T0 cutoff: 73rd percentile) | **27 (19–37)** | **12 (7–19)** | **<0.01** | −56 |
| **Ready-to-eat meals** | **n = 242** | **n = 223** | | |
| Any "high in" | 13 (9–18) | 9 (6–14) | 0.20 | −23 |
| High in energy (T0 cutoff: 91st percentile) | 6 (4–10) | 5 (3–9) | 0.71 | 0 |
| High in sugars (T0 cutoff: 99th percentile) | 0 | 0 | NA | NA |
| High in saturated fats (T0 cutoff: 94th percentile) | 5 (3–9) | 2 (0.7–5) | 0.08 | −60 |
| High in sodium (T0 cutoff: 90th percentile) | **10 (7–15)** | **5 (3–9)** | **0.03** | −50 |
| **Sausages** | **n = 362** | **n = 142** | | |
| Any "high in" | **81 (77–85)** | **32 (25–41)** | **<0.01** | −60 |
| High in energy (T0 cutoff: 85th percentile) | 15 (12–19) | 9 (5–15) | 0.09 | −40 |
| High in sugars (T0 cutoff: 99th percentile) | 0 | 0 | NA | NA |
| High in saturated fats (T0 cutoff: 46th percentile) | 17 (14–22) | 14 (9–21) | 0.35 | −18 |
| High in sodium (T0 cutoff: 23rd percentile) | **74 (69–78)** | **27 (20–35)** | **<0.01** | −64 |
| **Nonsausage meat products** | **n = 297** | **n = 101** | | |
| Any "high in" | 20 (15–25) | 25 (17–34) | 0.26 | +25 |
| High in energy (T0 cutoff: 98th percentile) | 0.1 (0.02–0.3) | 3 (0.6–9) | 0.16 | +900 |
| High in sugars (T0 cutoff: 99th percentile) | 0 | 0 | NA | NA |
| High in saturated fats (T0 cutoff: 81st percentile) | 16 (12–21) | 20 (13–29) | 0.41 | +25 |
| High in sodium (T0 cutoff: 95th percentile) | 4 (2–7) | 2 (0.2–7) | 0.27 | −50 |
| **Soups** | **n = 125** | **n = 69** | | |
| Any "high in" | **98 (94–100)** | **86 (75–93)** | **<0.01** | −12 |
| High in energy (T0 cutoff: 99th percentile) | 0 | 0 | NA | NA |

*(Continued)*

**Table 1.** (Continued)

| Food or beverage | T0, % (95% CI) | T1, % (95% CI) | *p*-value | Relative change, % of T0 |
|---|---|---|---|---|
| High in Sugars (T0 cutoff: 99th percentile) | 0 | 0 | NA | NA |
| High in saturated fats (T0 cutoff: 99th percentile) | 0 | 0 | NA | NA |
| High in sodium (T0 cutoff: 1st percentile) | **100** | **89 (79–96)** | **<0.01** | −11 |

Values represent the frequency and 95% CI of "high in" products.

Cutoffs correspond to the limits on the amount of energy or nutrient of concern for the initial implementation of the law (i.e., for solids, per 100g: 350 kcal of energy, 22.5 g of sugars, 6 g of saturated fats, 800 mg of sodium; for liquids, per 100 mL: 100 kcal of energy, 6 g of sugars, 3 g of saturated fats, 100 mg of sodium). The corresponding percentile was calculated according to T0 distribution of energy or nutrient of concern by food or beverage group.

Relative change: delta in the proportion between T0 and T1, relative to proportion in T0 (T0 − T1) × 100 ÷ T0; a negative sign represents a decrease, a positive sign represents an increase).

T0: preimplementation period, January to February 2015 + January to February 2016 (*n* = 4,055); T1: postimplementation period, January to February 2017 (*n* = 3,025). Comparison between T0 and T1 were done using Chi$^2$.

Significant *p*-values (i.e., <0.05) are bolded.

Note that discrepancies in the percentage of "high in" sodium and any "high in" among soups were given by differences in the denominator used in each case; the denominator for the former did not consider food items with missing information of sodium, whereas the latter did.

distribution below the initial cutoffs in the distribution of energy or the specific nutrients of concern in food and beverage groups listed above. Similar to the cross-sectional analysis, improvements in the proportion of "high in" products were also shown in the decrease of the

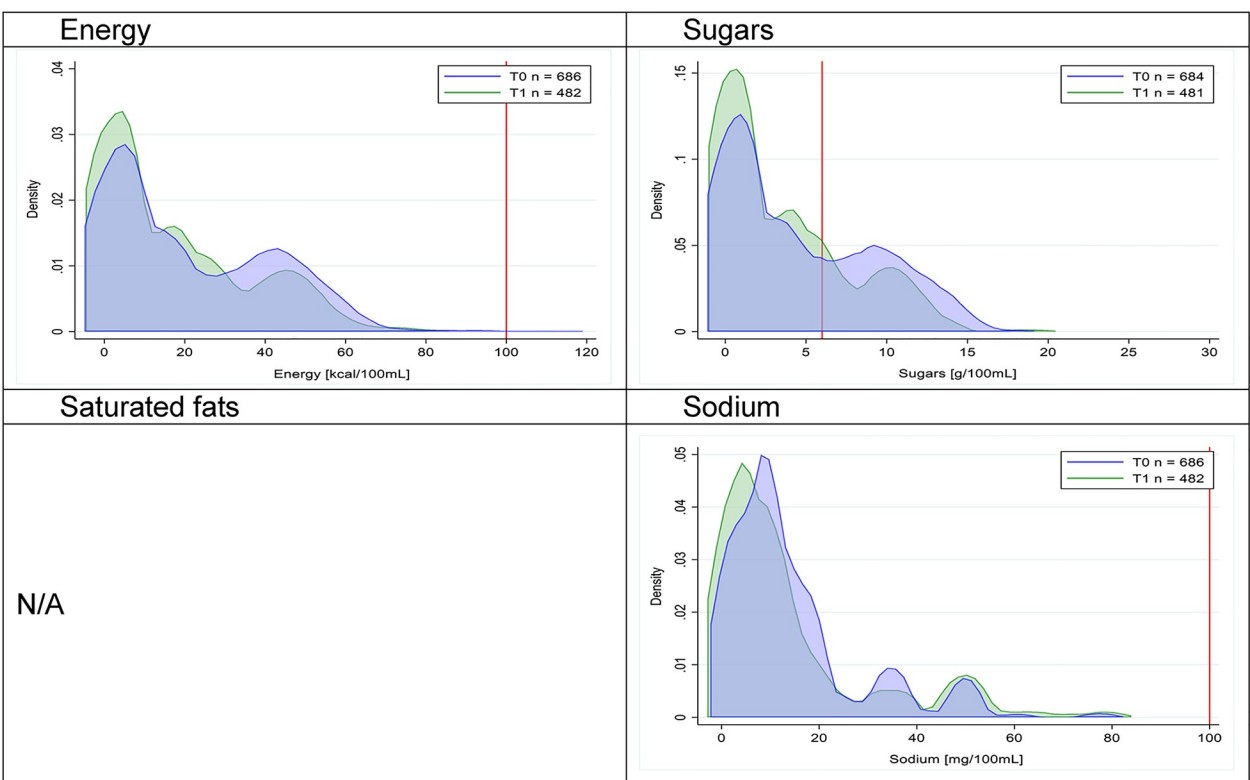

**Fig 2. Density curves for the amount of energy and nutrients of concern in beverages, cross-sectional samples.** The blue line represents the distribution in T0 (preimplementation), the green line represents the distribution in T1 (postimplementation), and the red line represents the cutoff for the amount of energy or nutrients of concern.

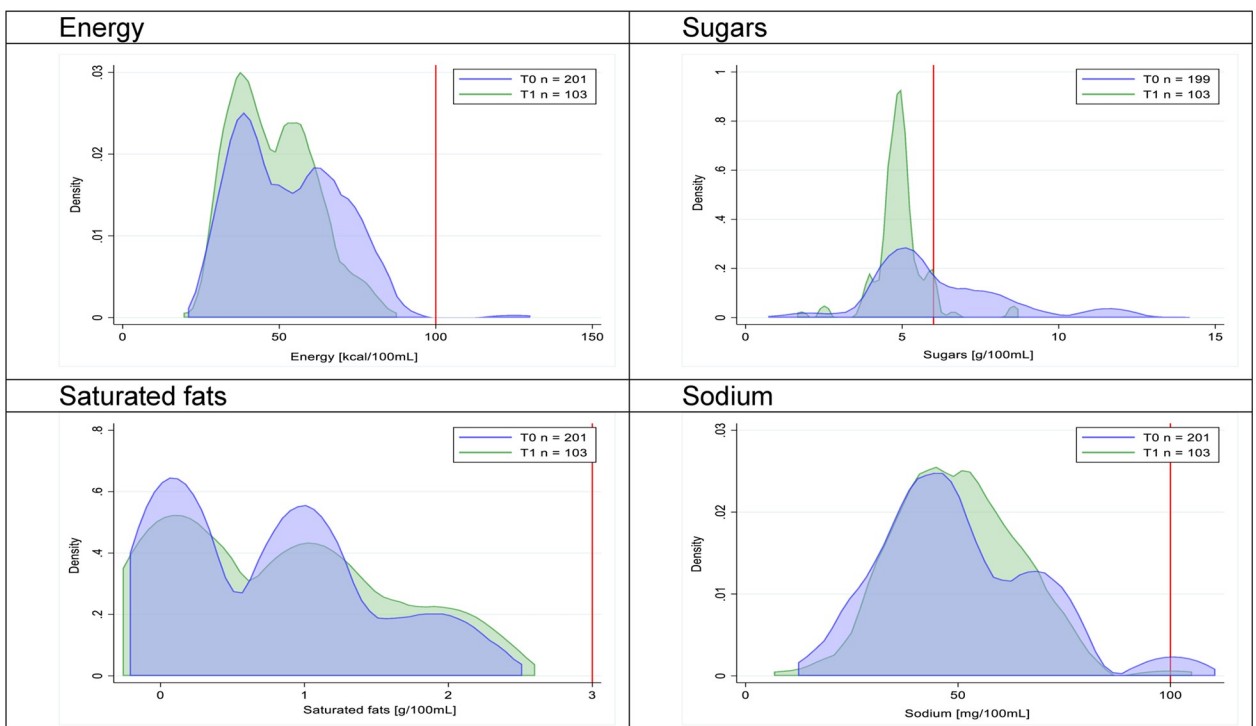

**Fig 3. Density curves for the amount of energy and nutrients of concern in milk and milk-based drinks, cross-sectional samples.** The blue line represents the distribution in T0 (preimplementation), the green line represents the distribution in T1 (postimplementation), and the red line represents the cutoff for the amount of energy or nutrients of concern.

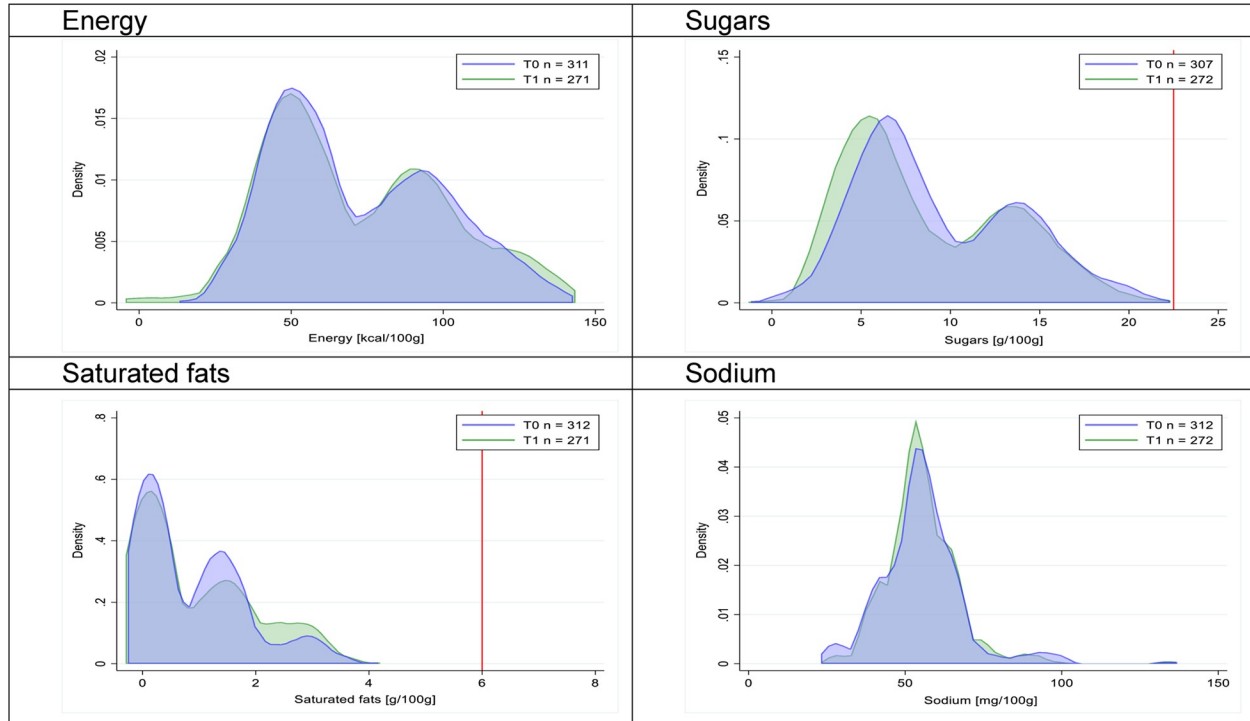

**Fig 4. Density curves for the amount of energy and nutrients of concern in yogurts, cross-sectional samples.** The blue line represents the distribution in T0 (preimplementation), the green line represents the distribution in T1 (postimplementation), and the red line represents the cutoff for the amount of energy or nutrients of concern.

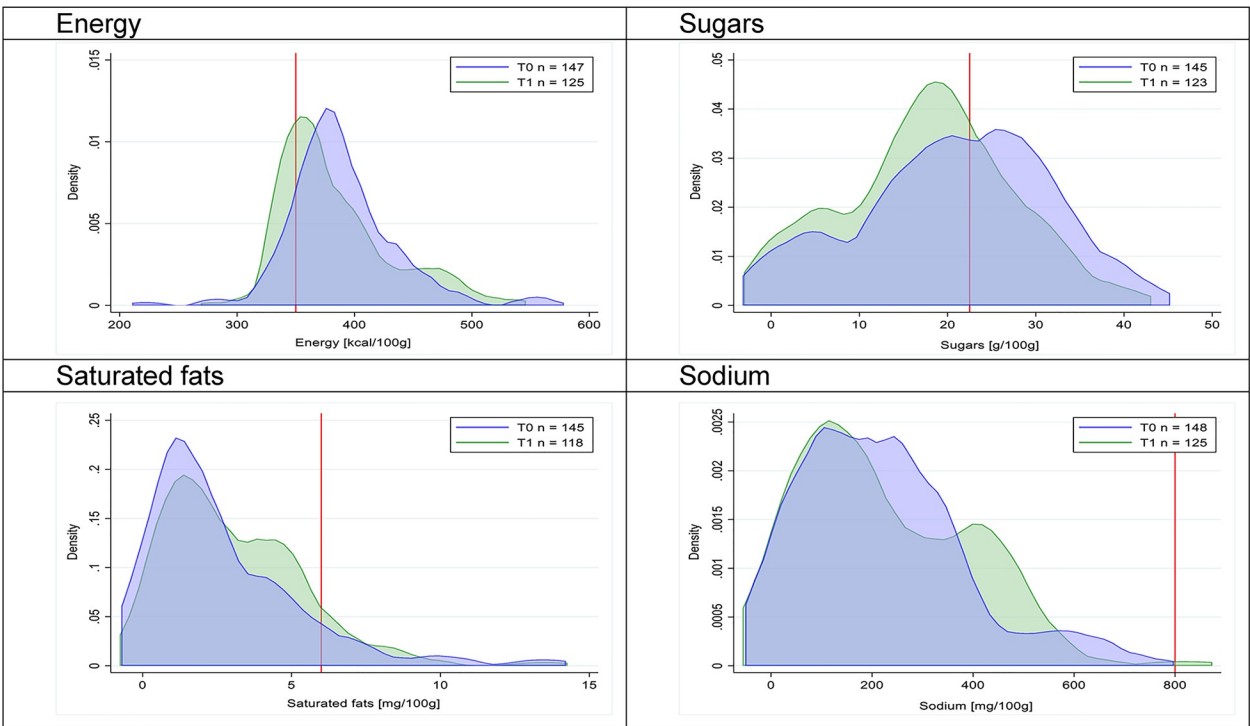

**Fig 5. Density curves for the amount of energy and nutrients of concern in breakfast cereals, cross-sectional samples.** The blue line represents the distribution in T0 (preimplementation), the green line represents the distribution in T1 (postimplementation), and the red line represents the cutoff for the amount of energy or nutrients of concern.

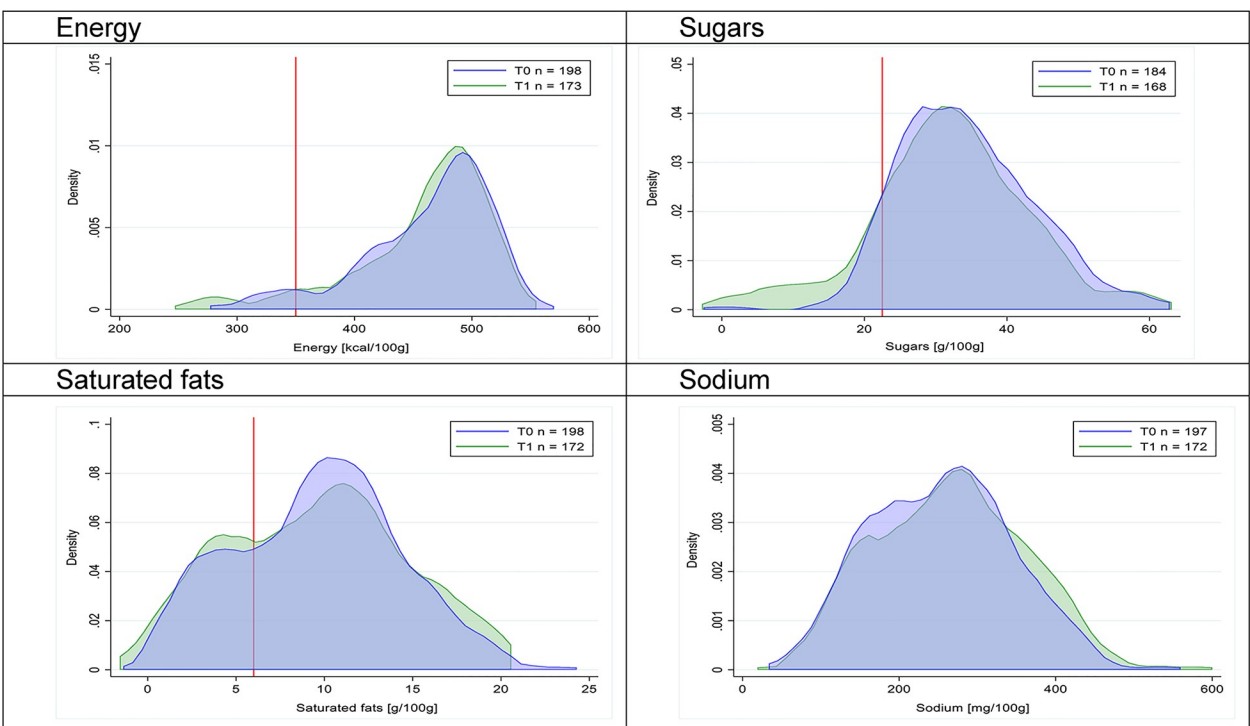

**Fig 6. Density curves for the amount of energy and nutrients of concern in sweet baked products, cross-sectional samples.** The blue line represents the distribution in T0 (preimplementation), the green line represents the distribution in T1 (postimplementation), and the red line represents the cutoff for the amount of energy or nutrients of concern.

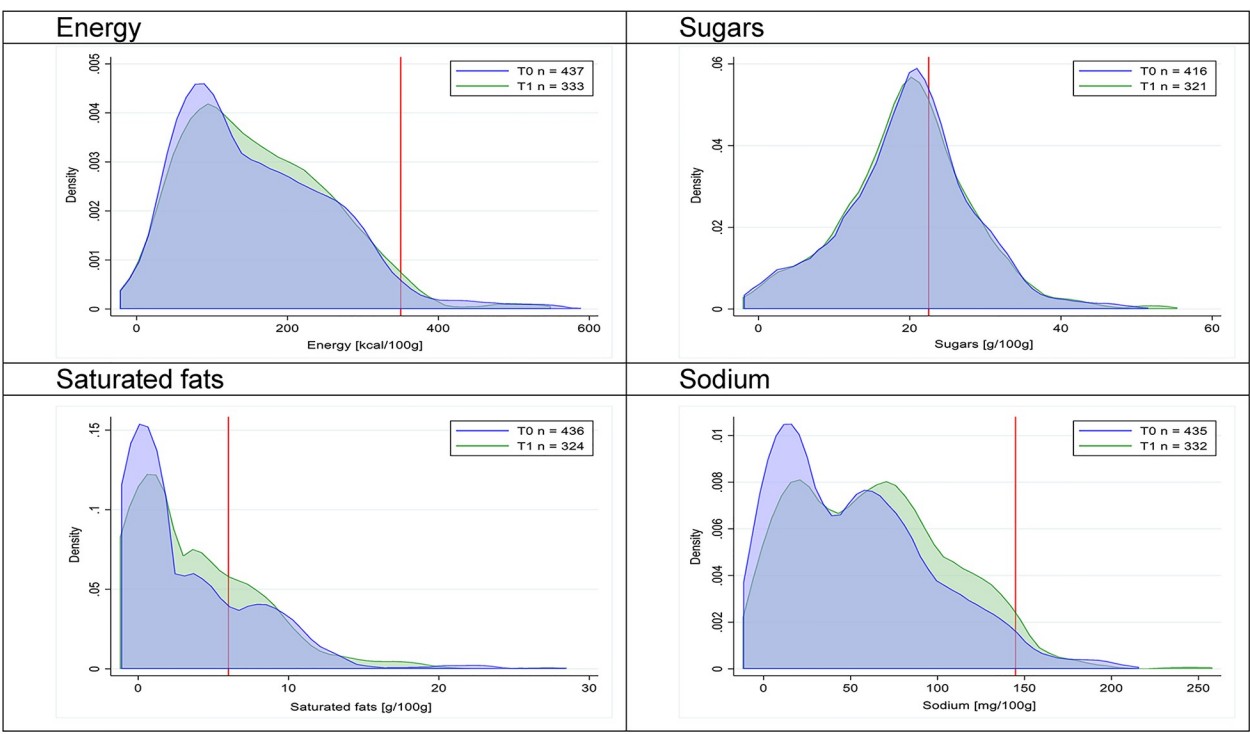

**Fig 7. Density curves for the amount of energy and nutrients of concern in desserts and ice creams, cross-sectional samples.** The blue line represents the distribution in T0 (preimplementation), the green line represents the distribution in T1 (postimplementation), and the red line represents the cutoff for the amount of energy or nutrients of concern.

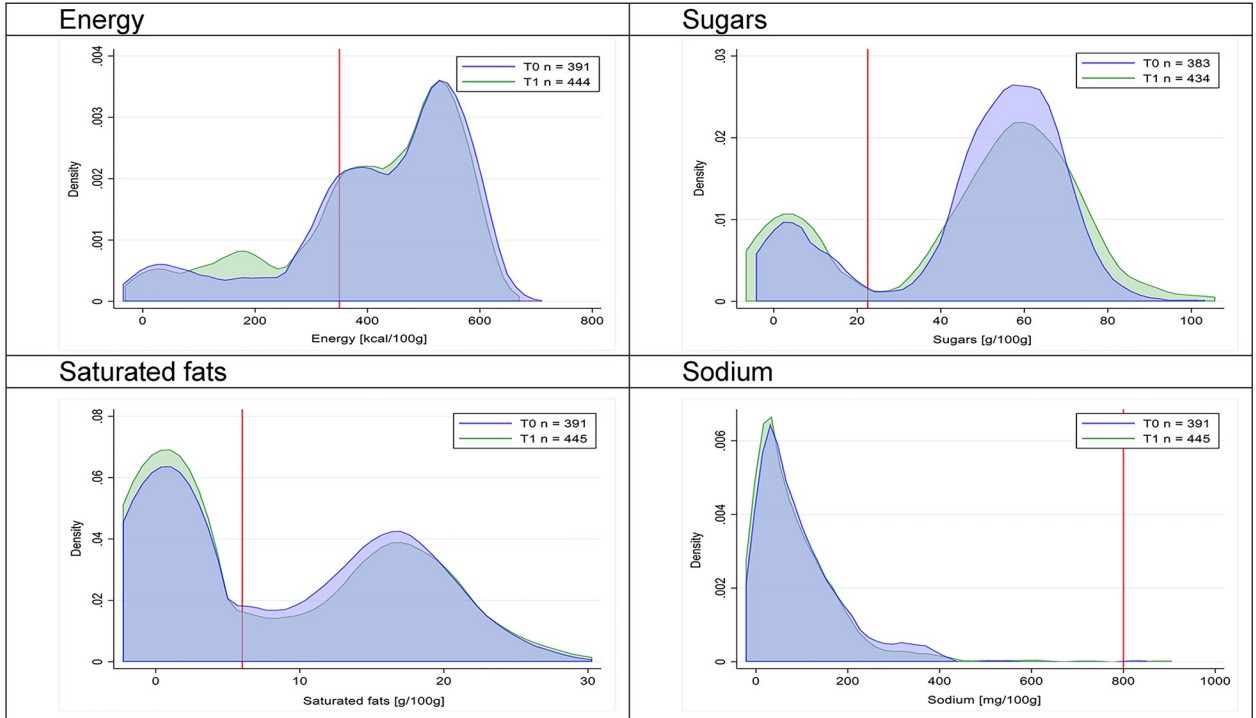

**Fig 8. Density curves for the amount of energy and nutrients of concern in candies and sweet confectioneries, cross-sectional samples.** The blue line represents the distribution in T0 (preimplementation), the green line represents the distribution in T1 (postimplementation), and the red line represents the cutoff for the amount of energy or nutrients of concern.

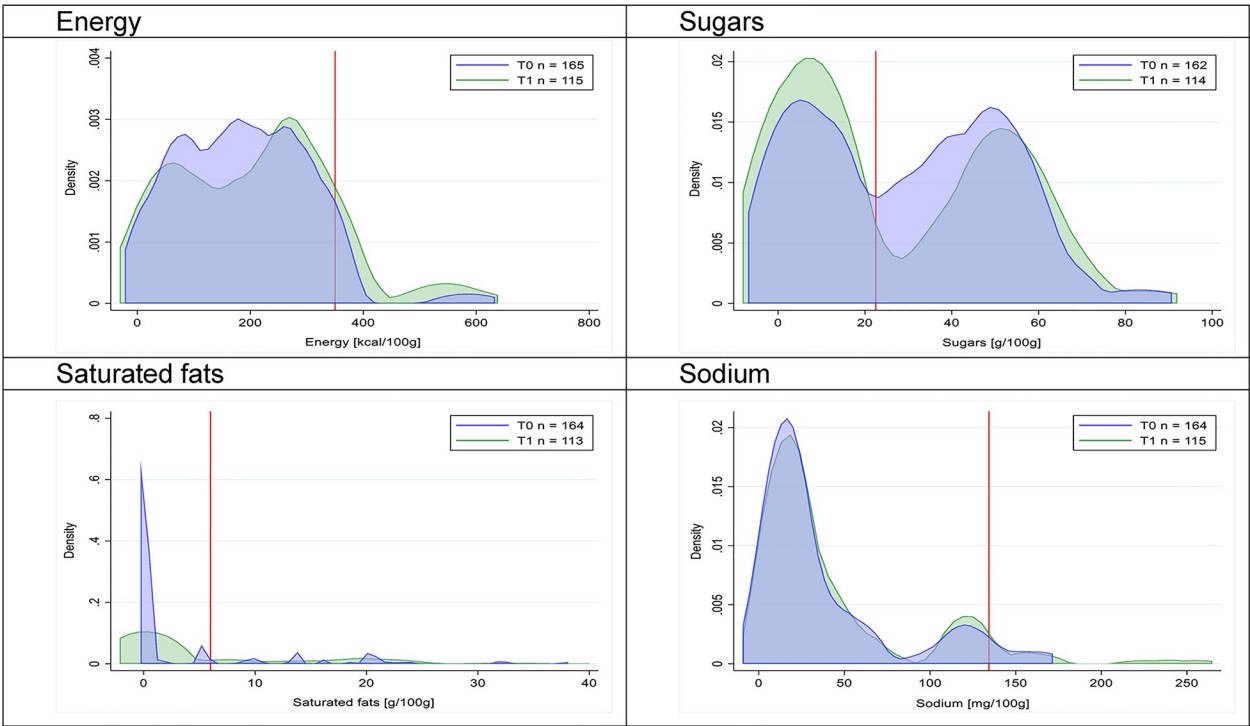

**Fig 9. Density curves for the amount of energy and nutrients of concern in sweet spreads, cross-sectional samples.** The blue line represents the distribution in T0 (preimplementation), the green line represents the distribution in T1 (postimplementation), and the red line represents the cutoff for the amount of energy or nutrients of concern.

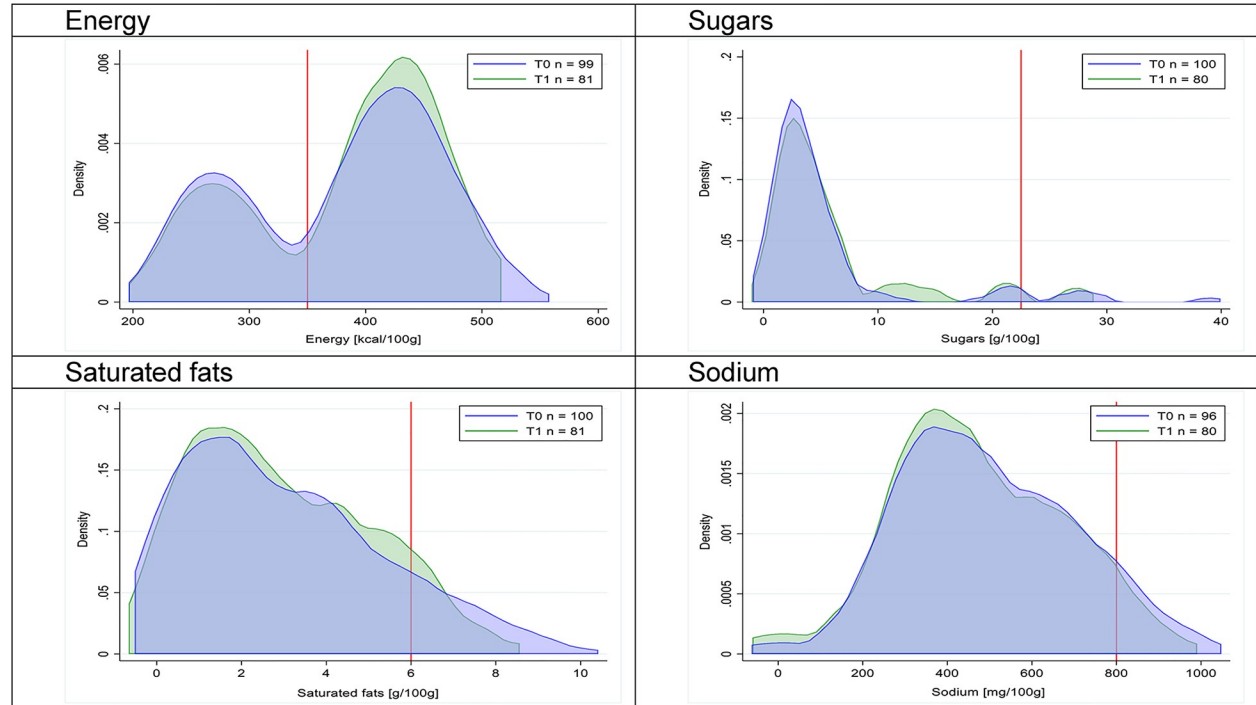

**Fig 10. Density curves for the amount of energy and nutrients of concern in savory baked products, cross-sectional samples.** The blue line represents the distribution in T0 (preimplementation), the green line represents the distribution in T1 (postimplementation), and the red line represents the cutoff for the amount of energy or nutrients of concern.

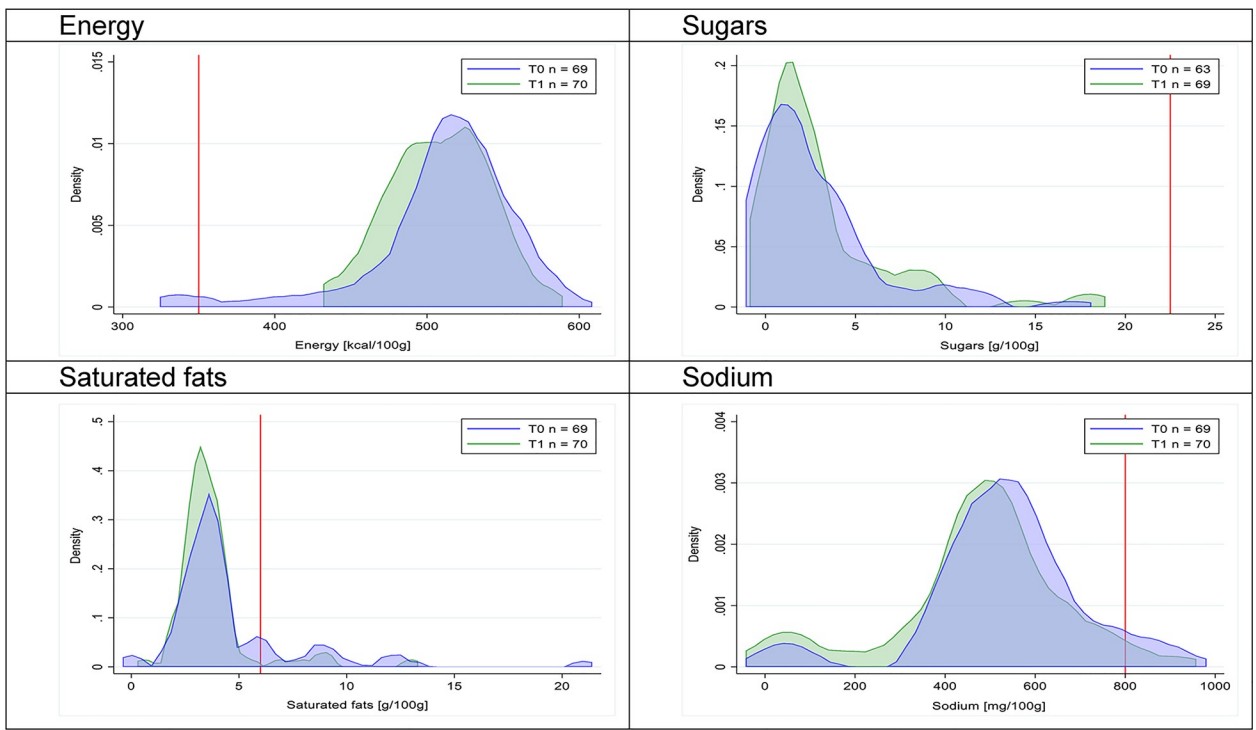

**Fig 11. Density curves for the amount of energy and nutrients of concern in savory snacks, cross-sectional samples.** The blue line represents the distribution in T0 (preimplementation), the green line represents the distribution in T1 (postimplementation), and the red line represents the cutoff for the amount of energy or nutrients of concern.

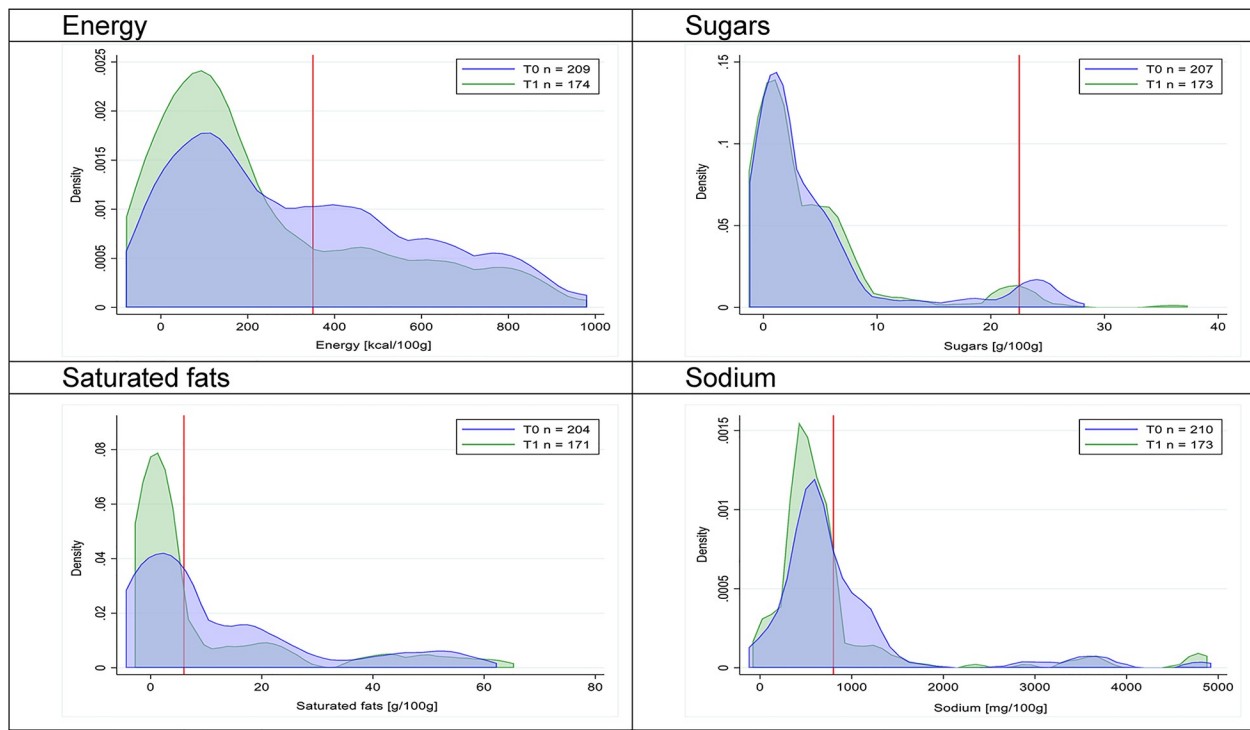

**Fig 12. Density curves for the amount of energy and nutrients of concern in savory spreads, cross-sectional samples.** The blue line represents the distribution in T0 (preimplementation), the green line represents the distribution in T1 (postimplementation), and the red line represents the cutoff for the amount of energy or nutrients of concern.

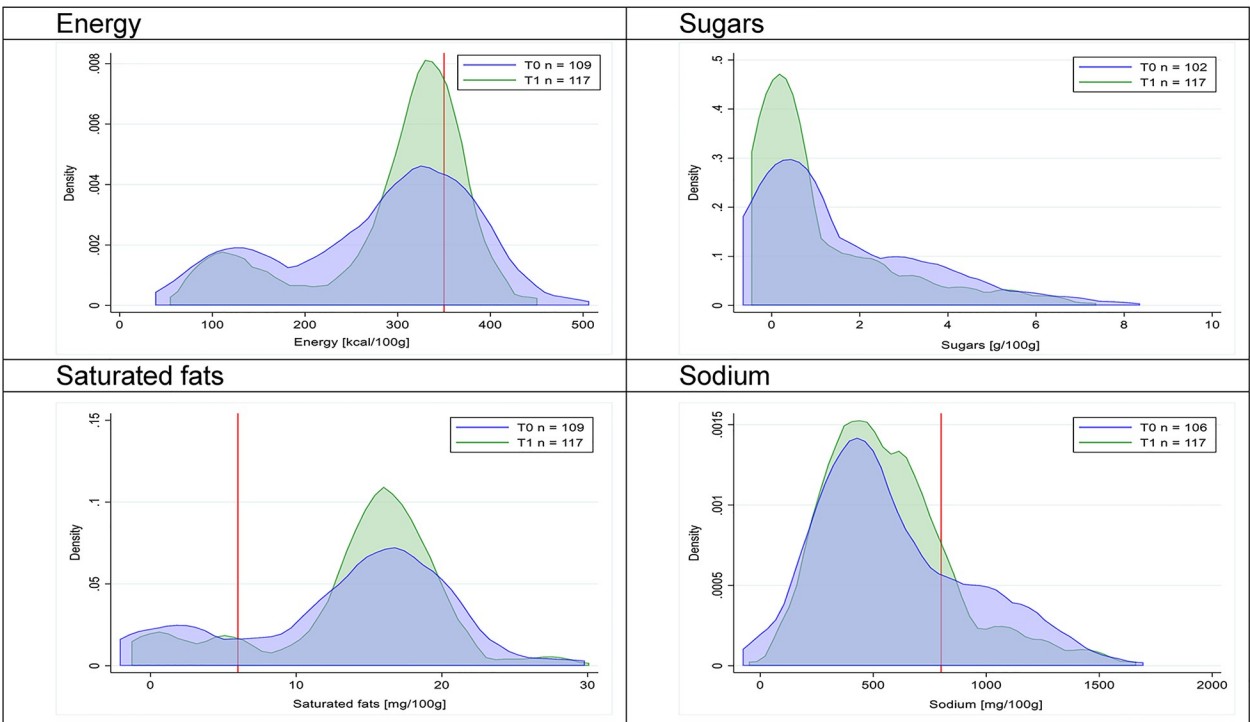

**Fig 13. Density curves for the amount of energy and nutrients of concern in cheeses, cross-sectional samples.** The blue line represents the distribution in T0 (preimplementation), the green line represents the distribution in T1 (postimplementation), and the red line represents the cutoff for the amount of energy or nutrients of concern.

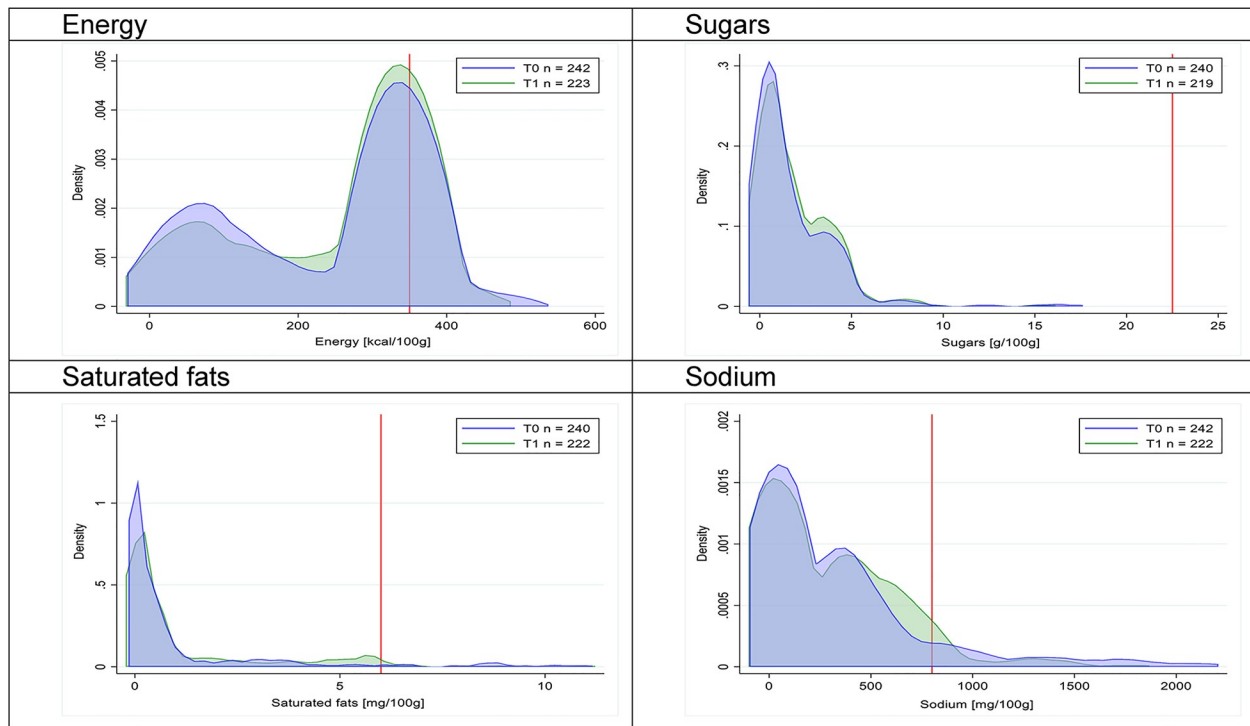

**Fig 14. Density curves for the amount of energy and nutrients of concern in ready-to-eat meals, cross-sectional samples.** The blue line represents the distribution in T0 (preimplementation), the green line represents the distribution in T1 (postimplementation), and the red line represents the cutoff for the amount of energy or nutrients of concern.

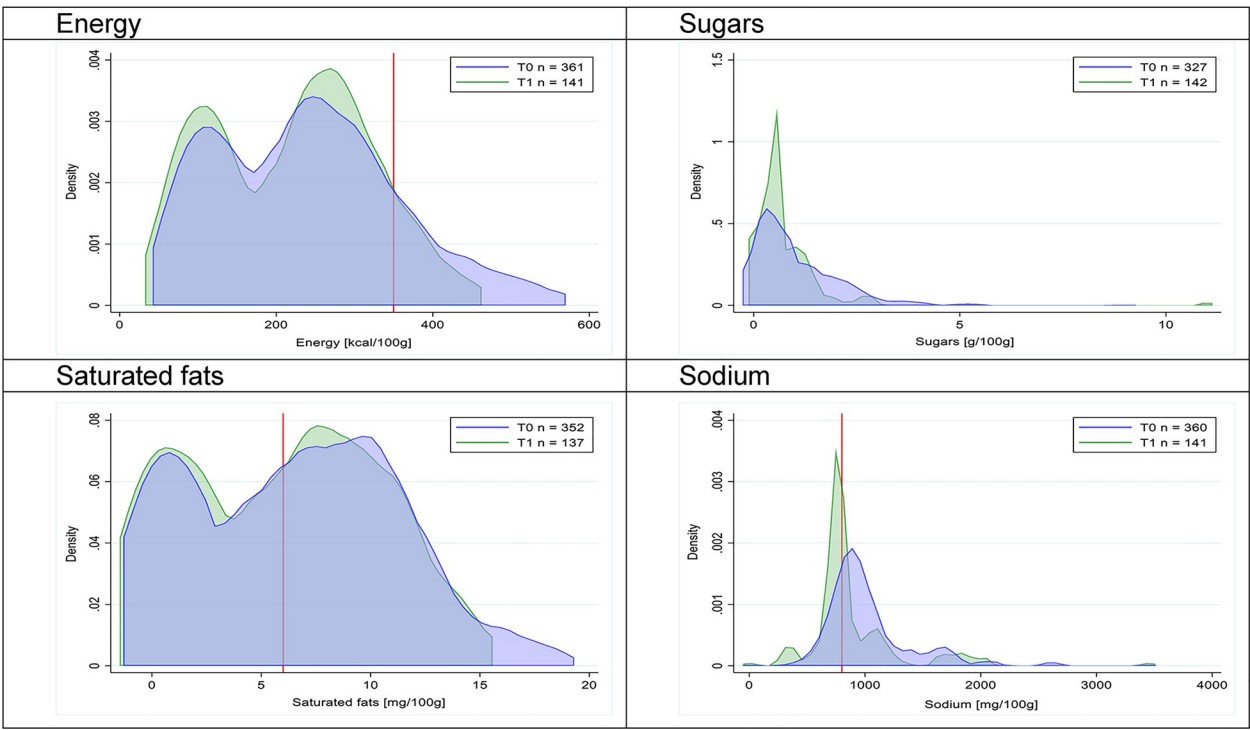

**Fig 15. Density curves for the amount of energy and nutrients of concern in sausages, cross-sectional samples.** The blue line represents the distribution in T0 (preimplementation), the green line represents the distribution in T1 (postimplementation), and the red line represents the cutoff for the amount of energy or nutrients of concern.

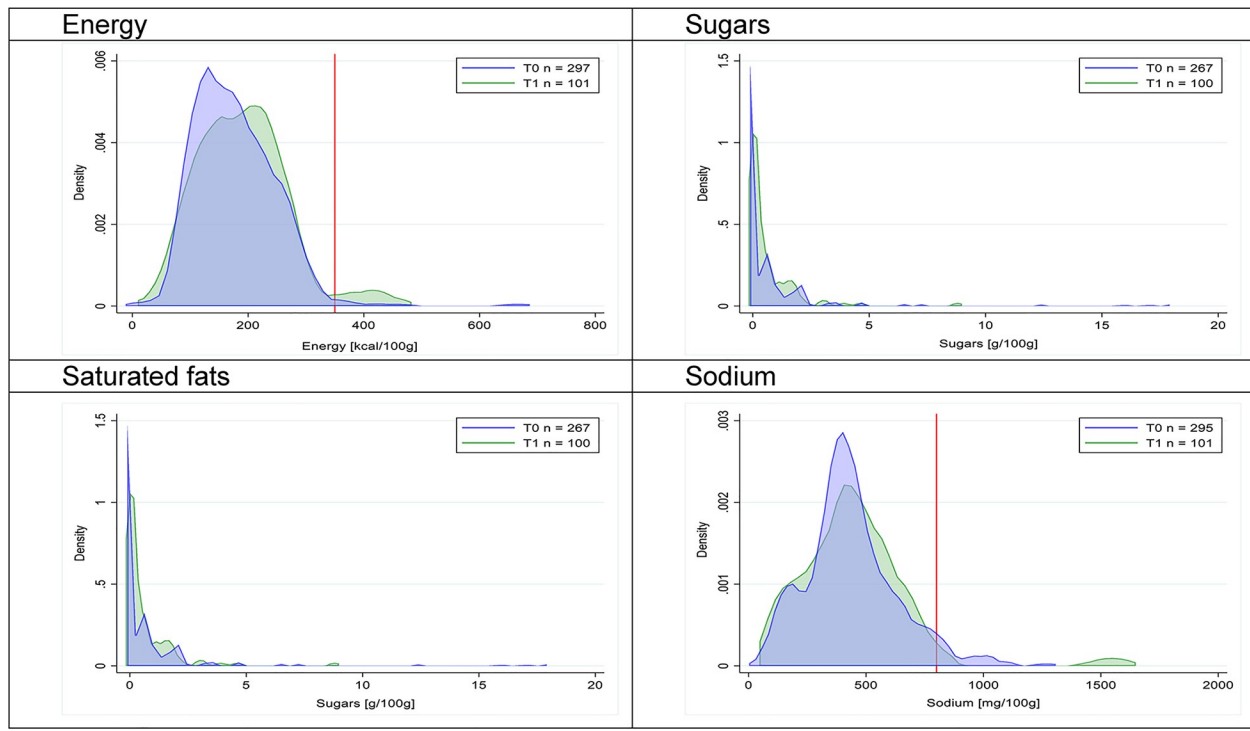

**Fig 16. Density curves for the amount of energy and nutrients of concern in nonsausage meat products, cross-sectional samples.** The blue line represents the distribution in T0 (preimplementation), the green line represents the distribution in T1 (postimplementation), and the red line represents the cutoff for the amount of energy or nutrients of concern.

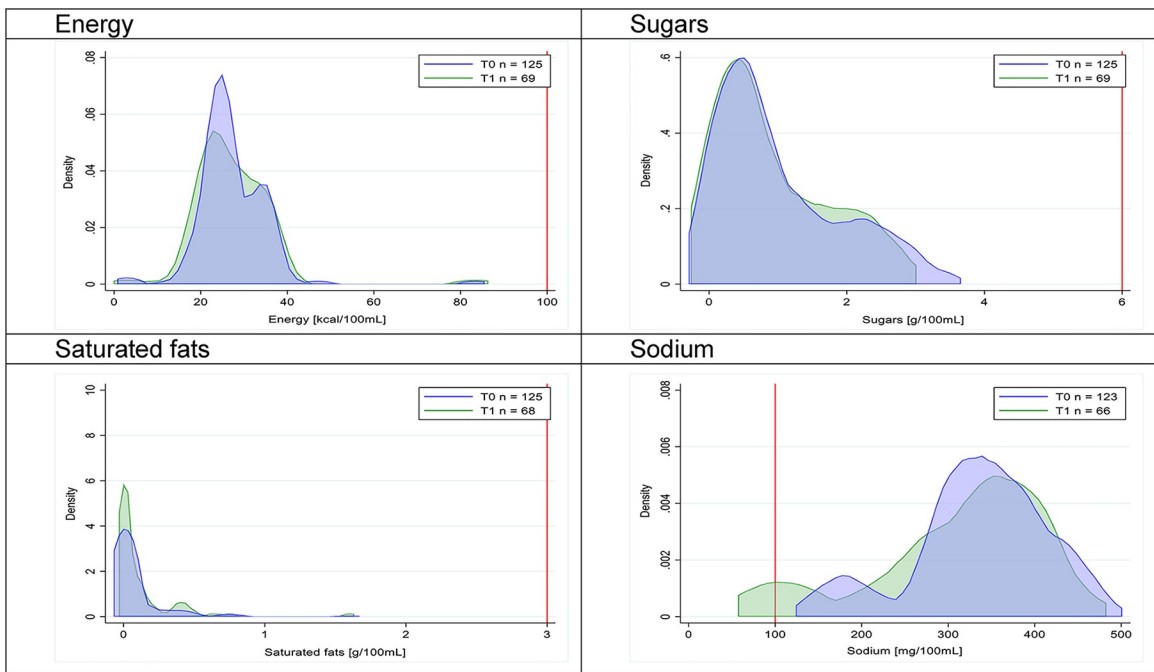

**Fig 17. Density curves for the amount of energy and nutrients of concern in soups, cross-sectional samples.** The blue line represents the distribution in T0 (preimplementation), the green line represents the distribution in T1 (postimplementation), and the red line represents the cutoff for the amount of energy or nutrients of concern.

quartile closest to the respective cutoff (i.e., 75th percentile of sugars in beverages [$p = 0.02$], 25th percentile of energy in breakfast cereals [$p = 0.03$], and 75th percentile of saturated fats in savory spreads [$p = 0.04$]). However, there were more exceptions than in the cross-sectional sample. On the other hand, there were some significant decreases in the amount of energy or nutrients of concern that were not associated with a decrease in the proportion of "high in" foods and beverages, as sugars and energy for yogurts, or saturated fats among savory spreads, among others (Table 4). In the longitudinal analyses, we did not confirm any of the right shift changes on energy or nutrient content distribution observed in the cross-sectional analyses.

## Discussion

To our knowledge, this is the first study to evaluate changes in the amount of energy and nutrients of concern in packaged foods and beverages available in the market after the initial implementation of the Chilean Law of Food Labelling and Advertising. Our results indicate that, in the cross-sectional analysis, compared to the preimplementation period, after <1 year of the law (i.e., June 2016 to January-February 2017) the proportion of any "high in" product decreased from 51% to 44%, mostly in food and beverage groups in which the regulatory cutoffs were below the 75th percentile of the nutrient or energy distribution. Decreases in the proportion of products "high in" were higher in sugars (6 out of the 16 food and beverage groups) and sodium (5 out of the 16 groups), whereas the proportion of "high in" saturated fats and "high in" energy decreased only in 1 and 2 food groups, respectively. In most cases, the energy and nutrient of concern distribution of the food and beverage food groups in which we observed decreases moved just below the regulatory cutoff. Several findings were confirmed in the longitudinal analysis.

**Table 2. Changes in quartiles of energy and nutrients of concern by food/beverages group; cross-sectional analysis.**

| Beverages and food | Energy (kcal/100g-mL) | | | Sugars (g/100g-mL) | | | Saturated fats (g/ 100 g-mL) | | | Sodium (mg/ 100 g-mL) | | |
|---|---|---|---|---|---|---|---|---|---|---|---|---|
| | T0 | T1 | *p*-value | T0 | T1 | *p*-value | T0 | T1 | *p*-value | T0 | T1 | *p*-value |
| **Beverages** | | | | | | | | | | | | |
| 25th percentile | **1.54** | **1.00** | **0.02** | 0.10 | 0.07 | 0.68 | - | - | - | **5.12** | **1.93** | **<0.01** |
| 50th percentile | 13.00 | 12.00 | 0.61 | 2.91 | 2.50 | 0.34 | - | - | - | **10.00** | **8.60** | **0.01** |
| 75th percentile | **38.00** | **26.00** | **<0.01** | **8.60** | **5.80** | **<0.01** | - | - | - | 18.00 | 17.00 | 0.38 |
| **Milks and milk-based drinks** | | | | | | | | | | | | |
| 25th percentile | 39.00 | 37.11 | 0.28 | 4.80 | 4.60 | 0.04 | **0.10** | **0.05** | **0.03** | 38.90 | 38.66 | 0.92 |
| 50th percentile | 52.58 | 46.00 | 0.05 | 5.41 | 4.90 | 0.03 | 0.90 | 0.90 | 1.00 | 46.39 | 49.00 | 0.28 |
| 75th percentile | **67.00** | **58.00** | **<0.01** | **7.30** | **5.10** | **<0.01** | 1.14 | 1.24 | 0.55 | 63.70 | 60.00 | 0.36 |
| **Yogurts** | | | | | | | | | | | | |
| 25th percentile | 50.00 | 49.00 | 0.32 | **5.80** | **5.00** | **<0.01** | 0.06 | 0.05 | 0.69 | 49.00 | 50.00 | 0.44 |
| 50th percentile | 65.00 | 60.00 | 0.33 | 8.00 | 7.10 | 0.11 | 0.48 | 0.42 | 0.74 | 55.00 | 55.00 | 1.00 |
| 75th percentile | 94.00 | 90.00 | 0.11 | 13.30 | 13.10 | 0.70 | **1.41** | **1.60** | **0.02** | 62.00 | 62.00 | 1.00 |
| **Breakfast cereals** | | | | | | | | | | | | |
| 25th percentile | **362.00** | **348.00** | **<0.01** | **15.00** | **10.00** | **0.03** | 1.10 | 1.30 | 0.33 | 101.0 | 103.0 | 0.92 |
| 50th percentile | **384.00** | **364.00** | **<0.01** | 21.00 | 18.00 | 0.06 | 1.90 | 2.40 | 0.20 | 195.00 | 200.00 | 0.86 |
| 75th percentile | 408.00 | 404.00 | 1.00 | **28.60** | **22.20** | **<0.01** | 3.90 | 4.30 | 0.37 | **320.0** | **395.00** | **<0.01** |
| **Sweet baked products** | | | | | | | | | | | | |
| 25th percentile | 428.00 | 437.00 | 0.43 | 27.80 | 25.30 | 0.05 | 6.30 | 5.60 | 0.37 | 182.00 | 188.00 | 0.73 |
| 50th percentile | 477.00 | 474.00 | 0.59 | 33.00 | 32.20 | 0.51 | 10.00 | 9.70 | 0.63 | 257.00 | 268.00 | 0.36 |
| 75th percentile | 501.00 | 500.00 | 0.82 | 40.00 | 38.20 | 0.35 | 12.40 | 12.90 | 0.51 | 311.00 | 331.00 | 0.17 |
| **Desserts and ice creams** | | | | | | | | | | | | |
| 25th percentile | 78.00 | 87.00 | 0.08 | 14.78 | 14.78 | 1.00 | 0.00 | 0.00 | - | **14.00** | **25.26** | **<0.01** |
| 50th percentile | 138.00 | 151.00 | 0.25 | 20.00 | 20.00 | 1.00 | **1.30** | **2.80** | **<0.01** | **48.00** | **63.64** | **<0.01** |
| 75th percentile | 229.09 | 223.64 | 0.68 | 24.55 | 25.45 | 0.30 | 5.74 | 6.00 | 0.73 | 80.00 | 90.91 | 0.06 |
| **Candies and sweet confectioneries** | | | | | | | | | | | | |
| 25th percentile | 343.00 | 341.00 | 0.91 | 43.00 | 32.00 | 0.14 | 0.00 | 0.00 | - | **24.00** | **17.00** | **0.03** |
| 50th percentile | 471.00 | 452.00 | 0.29 | 54.00 | 54.00 | 1.00 | 8.10 | 5.10 | 0.13 | 67.00 | 58.00 | 0.23 |
| 75th percentile | 544.00 | 533.00 | 0.06 | 64.00 | 66.00 | 0.06 | 16.80 | 17.00 | 0.75 | 137.00 | 130.00 | 0.56 |
| **Sweet spreads** | | | | | | | | | | | | |
| 25th percentile | 105.00 | 81.00 | 0.19 | 5.63 | 4.30 | 0.25 | 0.00 | 0.00 | - | 12.00 | 13.00 | 0.52 |
| 50th percentile | 197.00 | 210.00 | 0.62 | **28.00** | **10.00** | **0.02** | 0.00 | 0.00 | - | 20.00 | 20.00 | 1.00 |
| 75th percentile | **250.00** | **308.00** | **<0.01** | 48.90 | 49.20 | 0.90 | **0.42** | **8.32** | **0.02** | 52.50 | 51.00 | 0.94 |
| **Savory baked products** | | | | | | | | | | | | |
| 25th percentile | 283.00 | 336.00 | 0.13 | 2.00 | 2.40 | 0.26 | 0.98 | 1.20 | 0.46 | 332.00 | 335.00 | 0.93 |
| 50th percentile | 407.00 | 413.00 | 0.67 | 3.30 | 3.70 | 0.47 | 2.90 | 3.00 | 0.84 | 472.00 | 463.00 | 0.84 |
| 75th percentile | 447.00 | 437.00 | 0.44 | 5.50 | 6.20 | 0.70 | 4.80 | 5.00 | 0.76 | 640.00 | 661.00 | 0.68 |
| **Savory snacks** | | | | | | | | | | | | |
| 25th percentile | 499.00 | 483.00 | 0.05 | 0.20 | 0.67 | 0.13 | 2.90 | 2.90 | 1.00 | 452.00 | 404.00 | 0.31 |
| 50th percentile | 519.00 | 508.00 | 0.13 | 1.72 | 2.00 | 0.62 | **3.90** | **3.31** | **0.01** | 538.00 | 492.00 | 0.06 |
| 75th percentile | 543.00 | 532.00 | 0.11 | 4.00 | 3.70 | 0.81 | 4.30 | 4.00 | 0.69 | 608.00 | 590.00 | 0.70 |
| **Savory spreads** | | | | | | | | | | | | |
| 25th percentile | **88.00** | **50.00** | **<0.01** | 0.30 | 0.24 | 0.52 | 0.00 | 0.00 | - | **470.00** | **399.00** | **0.03** |
| 50th percentile | **253.00** | **114.00** | **<0.01** | 2.00 | 1.90 | 0.88 | **3.70** | **0.07** | **0.02** | **653.00** | **511.00** | **<0.01** |
| 75th percentile | **516.00** | **359.00** | **0.03** | 5.50 | 5.70 | 0.77 | **18.20** | **10.70** | **0.04** | **1,070.53** | **745.00** | **<0.01** |
| **Cheeses** | | | | | | | | | | | | |
| 25th percentile | 228.00 | 273.00 | 0.21 | 0.00 | 0.00 | - | 10.94 | 12.80 | 0.48 | 364.00 | 361.00 | 0.91 |

*(Continued)*

**Table 2.** (Continued)

| Beverages and food | Energy (kcal/100g-mL) | | | Sugars (g/100g-mL) | | | Saturated fats (g/ 100 g-mL) | | | Sodium (mg/ 100 g-mL) | | |
|---|---|---|---|---|---|---|---|---|---|---|---|---|
| | T0 | T1 | *p*-value | T0 | T1 | *p*-value | T0 | T1 | *p*-value | T0 | T1 | *p*-value |
| 50th percentile | 305.00 | 320.00 | 0.23 | 0.70 | 0.20 | 0.06 | 14.60 | 15.40 | 0.31 | 492.40 | 524.30 | 0.54 |
| 75th percentile | 350.00 | 346.00 | 0.60 | 2.60 | 1.80 | 0.21 | 18.34 | 17.92 | 0.56 | 842.00 | 718.00 | 0.11 |
| *Ready-to-eat meals* | | | | | | | | | | | | |
| 25th percentile | 114.00 | 136.00 | 0.47 | 0.00 | 0.20 | 0.09 | 0.00 | 0.00 | - | **9.70** | **5.00** | **<0.01** |
| 50th percentile | 329.00 | 336.00 | 0.69 | 0.91 | 1.20 | 0.14 | **0.00** | **0.30** | **<0.01** | 191.00 | 241.00 | 0.39 |
| 75th percentile | 342.00 | 342.00 | 1.00 | 2.80 | 2.90 | 0.83 | 0.60 | 0.90 | 0.47 | 430.00 | 497.00 | 0.21 |
| *Sausages* | | | | | | | | | | | | |
| 25th percentile | 133.00 | 116.00 | 0.37 | **0.22** | **0.50** | **0.01** | 1.85 | 0.80 | 0.22 | **800.00** | **730.00** | **<0.01** |
| 50th percentile | 236.00 | 235.00 | 0.95 | 0.50 | 0.50 | 1.00 | 6.50 | 5.99 | 0.47 | **929.00** | **786.00** | **<0.01** |
| 75th percentile | **320.00** | **283.00** | **0.03** | **1.50** | **0.90** | **<0.01** | 10.00 | 9.30 | 0.17 | **1,120.00** | **952.00** | **0.03** |
| *Nonsausage meat products* | | | | | | | | | | | | |
| 25th percentile | 123.00 | 135.00 | 0.20 | 0.00 | 0.00 | - | 1.00 | 1.51 | 0.11 | 333.00 | 320.00 | 0.72 |
| 50th percentile | **166.00** | **188.00** | **0.05** | **0.00** | **0.10** | **0.02** | **2.10** | **3.10** | **0.03** | 410.00 | 429.00 | 0.39 |
| 75th percentile | 220.00 | 238.00 | 0.17 | 0.50 | 0.80 | 0.27 | 4.70 | 5.90 | 0.13 | 522.00 | 560.00 | 0.31 |
| *Soups* | | | | | | | | | | | | |
| 25th percentile | 24.00 | 22.00 | 0.08 | 0.35 | 0.25 | 0.24 | 0.00 | 0.00 | - | 298.17 | 270.00 | 0.18 |
| 50th percentile | 25.50 | 25.00 | 0.66 | 0.75 | 0.58 | 0.28 | 0.00 | 0.00 | 1.00 | 334.5 | 326.5 | 0.60 |
| 75th percentile | 32.40 | 32.90 | 0.81 | 1.52 | 1.51 | 0.98 | 0.00 | 0.11 | 0.08 | 381.27 | 376.67 | 0.70 |

Quartiles and *p*-values were obtained from quantile regressions models (one model per nutrient per food or beverage group), using implementation period as independent variable (T0 = 0, T1 = 1).

Significant *p*-values (i.e., <0.05) are bolded.

T0: preimplementation period, January to February 2015 + January to February 2016 (*n* = 4,055).

T1: postimplementation period, January to February 2017 (*n* = 3,025).

Although food and beverage reformulation has been suggested as a key strategy for obesity prevention, there is scarce evidence on how real-life initiatives can encourage reformulation. In Australia and New Zealand, significant improvements were seen in the amount of energy (1.5% decrease) and sodium (6.7% decrease) of products that adopted the Health Star Ratings (HSR) FOP label, and reformulation may have been even higher among food products targeted to children [18, 19]. However, HSR was a voluntary initiative implemented by less than 5% of local food suppliers. The nutritional quality of children's menus in fast food restaurants was reported to have improved after the implementation of an ordinance prohibiting toy incentives to children together with food of low nutritional quality in one county of California, United States [29]. There are also reports of improvement in the nutritional quality of foods driven by other kinds of voluntary actions [9, 10, 16, 30–33]. However, the overall impact of these initiatives on the whole food supply has not been well characterized.

In the Chilean law, regulatory cutoffs were defined based on natural foods and liquids considered gold standards of a healthy diet [24]. Therefore, the same cutoffs were used for all food and beverage groups, only considering differences for liquids and solids. Regulatory policies that are more oriented to promoting reformulation might prefer considering specific cutoffs for each of the food and beverage groups [34]. Our results suggest that setting up cutoffs that are below the 75th percentile of the nutrient distribution would allow to achieve the change looked for; conversely, cutoffs defined on the top end of the distribution would not promote significant changes on the food supply. The Chilean law had 2 other phases of implementation in which regulatory cutoffs became increasingly stricter; therefore, further analyses of changes

**Table 3. Changes between T0 and T1 in the proportion of "high in" energy and nutrients of concern (or any "high in") by food or beverage group, longitudinal analysis.**

| Beverages and foods | T0, % (95% CI) | T1, % (95% CI) | p-value | Relative change, % of T0 |
|---|---|---|---|---|
| **Beverages,** n = 326 | | | | |
| Any "high in" | **20 (16–25)** | **9 (6–12)** | **<0.01** | −55 |
| High in energy (T0 cutoff: 99th percentile) | 0 | 0 | - | NA |
| High in sugars (T0 cutoff: 69th percentile) | **20 (16–25)** | **9 (6–12)** | **<0.01** | −55 |
| High in saturated fats (T0 cutoff: NA) | 0 | 0 | - | NA |
| High in sodium (T0 cutoff: 99th percentile) | 0 | 0 | - | NA |
| **Milks and milk-based drinks,** n = 76 | | | | |
| Any "high in" | **30 (21–42)** | **0** | **<0.01** | −100 |
| High in energy (T0 cutoff: 98th percentile) | 1 (0.03–7) | 0 | 0.33 | −100 |
| High in sugars (T0 cutoff: 63rd percentile) | **28 (18–39)** | **0** | **<0.01** | −100 |
| High in saturated fats (T0 cutoff: 99th percentile) | 0 | 0 | - | NA |
| High in sodium (T0 cutoff: 98th percentile) | 1 (0.03–7) | 0 | 0.32 | NA |
| **Yogurts,** *n* = 181 | | | | |
| Any "high in" | 0 | 0 | - | NA |
| High in energy (T0 cutoff: 99th percentile) | 0 | 0 | - | NA |
| High in sugars (T0 cutoff: 99th percentile) | 0 | 0 | - | NA |
| High in saturated fats (T0 cutoff: 99th percentile) | 0 | 0 | - | NA |
| High in sodium (T0 cutoff: 99th percentile) | 0 | 0 | - | NA |
| **Breakfast cereals,** *n* = 67 | | | | |
| Any "high in" | **78 (66–87)** | **55 (43–67)** | **<0.01** | −29 |
| High in energy (T0 cutoff: 12th percentile) | **78 (66–87)** | **55 (44–67)** | **<0.01** | −29 |
| High in sugars (T0 cutoff: 57th percentile) | **42 (30–54)** | **20 (11–32)** | **<0.01** | −52 |
| High in saturated fats (T0 cutoff: 87th percentile) | 8 (3–17) | 6 (2–15) | 0.75 | −25 |
| High in sodium (T0 cutoff: 99th percentile) | 0 | 0 | - | NA |
| **Sweet baked products,** *n* = 118 | | | | |
| Any "high in" | 100 | 99 (95–100) | 0.32 | −1 |
| High in energy (T0 cutoff: 4th percentile) | 96 (90–99) | 97 (93–99) | 0.32 | −1 |
| High in sugars (T0 cutoff: 6th percentile) | 94 (88–97) | 89 (81–94) | 0.06 | −5 |
| High in saturated fats (T0 cutoff: 19th percentile) | 78 (69–85) | 73 (64–81) | 0.08 | −6 |
| High in sodium (T0 cutoff: 99th percentile) | 0 | 0 | - | NA |
| **Desserts and ice creams,** *n* = 230 | | | | |
| Any "high in" | **45 (39–52)** | **38 (32–44)** | **<0.01** | −16 |
| High in energy (T0 cutoff: 97th percentile) | 2 (0.1–5) | 3 (0.1–6) | 0.48 | +50 |
| High in sugars (T0 cutoff: 62nd percentile) | **37 (31–44)** | **30 (24–37)** | **<0.01** | −19 |
| High in saturated fats (T0 cutoff: 73rd percentile) | 27 (21–33) | 25 (19–31) | 0.05 | −7 |
| High in sodium (T0 cutoff: 99th percentile) | 0 | 0 | - | NA |
| **Candies and sweet confectioneries,** *n* = 216 | | | | |
| Any "high in" | 88 (83–92) | 88 (83–92) | 0.56 | 0 |
| High in energy (T0 cutoff: 24th percentile) | 75 (69–81) | 75 (69–81) | 1.00 | 0 |
| High in sugars (T0 cutoff: 18th percentile) | 82 (76–87) | 81 (75–86) | 1.00 | −1 |
| High in saturated fats (T0 cutoff: 46th percentile) | **54 (47–60)** | **50 (43–57)** | **<0.01** | −7 |
| High in sodium (T0 cutoff: 99th percentile) | 0 | 0 | - | NA |
| **Sweet spreads,** *n* = 73 | | | | |
| Any "high in" | 79 (68–88) | 71 (59–81) | 0.06 | −10 |
| High in energy (T0 cutoff: 97th percentile) | 3 (0.3–10) | 4 (0.9–12) | 0.32 | +33 |
| High in sugars (T0 cutoff: 45th percentile) | **56 (44–68)** | **45 (34–57)** | **<0.01** | −20 |

(*Continued*)

**Table 3.** (Continued)

| Beverages and foods | T0, % (95% CI) | T1, % (95% CI) | *p*-value | Relative change, % of T0 |
|---|---|---|---|---|
| High in saturated fats (T0 cutoff: 70th percentile) | 28 (18–40) | 30 (19–42) | 1.00 | +7 |
| High in sodium (T0 cutoff: 99th percentile) | 0 | 0 | - | NA |
| **Savory baked products,** *n* = 61 | | | | |
| Any "high in" | 62 (49–74) | 62 (49–74) | - | 0 |
| High in energy (T0 cutoff: 32nd percentile) | 62 (49–74) | 62 (49–74) | - | 0 |
| High in sugars (T0 cutoff: 96th percentile) | 3 (0.3–11) | 2 (0.4–9) | 0.32 | 0 |
| High in saturated fats (T0 cutoff: 87th percentile) | 11 (5–22) | 5 (1–14) | 0.05 | −55 |
| High in sodium (T0 cutoff: 94th percentile) | 5 (1–14) | 3 (0.4–12) | 0.32 | −40 |
| **Savory snacks,** *n* = 29 | | | | |
| Any "high in" | 90 (73–98) | 100 | 0.08 | +11 |
| High in energy (T0 cutoff: 6th percentile) | 90 (73–98) | 97 (82–100) | 0.16 | +8 |
| High in sugars (T0 cutoff: 99th percentile) | 0 | 0 | - | NA |
| High in saturated fats (T0 cutoff: 83rd percentile) | 14 (4–32) | 10 (2–27) | 0.56 | −29 |
| High in sodium (T0 cutoff: 93rd percentile) | 7 (0.1–23) | 3 (0.01–18) | 0.32 | −57 |
| **Savory spreads,** *n* = 112 | | | | |
| Any "high in" | **72 (62–80)** | **55 (46–65)** | **<0.01** | −24 |
| High in energy (T0 cutoff: 55th percentile) | **43 (34–53)** | **34 (25–43)** | **<0.01** | −21 |
| High in sugars (T0 cutoff: 95th percentile) | **4 (1–10)** | **0** | **0.03** | −100 |
| High in saturated fats (T0 cutoff: 52nd percentile) | **48 (38–57)** | **35 (27–45)** | **<0.01** | −27 |
| High in sodium (T0 cutoff: 71st percentile) | **29 (20–38)** | **17 (11–25)** | **<0.01** | −41 |
| **Cheeses,** *n* = 60 | | | | |
| Any "high in" | 82 (70–90) | 82 (70–90) | - | 0 |
| High in energy (T0 cutoff: 78th percentile) | 22 (12–34) | 20 (11–32) | 0.71 | −9 |
| High in sugars (T0 cutoff: 99 percentile) | 0 | 0 | - | NA |
| High in saturated fats (T0 cutoff: 20th percentile) | 80 (68–89) | 80 (68–89) | - | 0 |
| High in sodium (T0 cutoff: 67th percentile) | **32 (20–45)** | **18 (10–30)** | **<0.01** | −44 |
| **Ready-to-eat meals,** *n* = 109 | | | | |
| Any "high in" | 16 (9–24) | 13 (7–21) | 0. 56 | −19 |
| High in energy (T0 cutoff: 91st percentile) | 8 (4–15) | 7 (3–14) | 0.80 | −13 |
| High in sugars (T0 cutoff: 99th percentile) | 0 | 0 | - | NA |
| High in saturated fats (T0 cutoff: 94th percentile) | 6 (2–12) | 3 (0.1–8) | 0.29 | −50 |
| High in sodium (T0 cutoff: 89th percentile) | 10 (5–17) | 7 (3–14) | 0.48 | −30 |
| **Sausages,** *n* = 120 | | | | |
| Any "high in" | **81 (73–87)** | **31 (23–40)** | **<0.01** | −62 |
| High in energy (T0 cutoff: 90th percentile) | 9 (5–16) | 8 (4–15) | 0.56 | −11 |
| High in sugars (T0 cutoff: 99th percentile) | 0 | 0 | - | NA |
| High in saturated fats (T0 cutoff: 52nd percentile) | 12 (7–19) | 12 (7–19) | 1.00 | 0 |
| High in sodium (T0 cutoff: 23rd percentile) | **73 (64–81)** | **27 (19–36)** | **<0.01** | −63 |
| **Nonsausage meat products,** *n* = 77 | | | | |
| Any "high in" | 31 (21–43) | 27 (18–39) | 0.37 | −13 |
| High in energy (T0 cutoff: 97th percentile) | 3 (0.3–9) | 3 (0.3–9) | - | 0 |
| High in sugars (T0 cutoff: 99th percentile) | 0 | 0 | - | NA |
| High in saturated fats (T0 cutoff: 69th percentile) | 28 (18–39) | 22 (13–33) | 0.21 | −21 |
| High in sodium (T0 cutoff: 94th percentile) | 5 (1–13) | 3 (0.3–9) | 0.16 | −40 |
| **Soups,** *n* = 57 | | | | |
| Any "high in" | 96 (88–100) | 95 (85–99) | 0.65 | −1 |
| High in energy (T0 cutoff: 99th percentile) | 0 | 0 | - | NA |

(*Continued*)

**Table 3.** (Continued)

| Beverages and foods | T0, % (95% CI) | T1, % (95% CI) | *p*-value | Relative change, % of T0 |
|---|---|---|---|---|
| High in sugars (T0 cutoff: 99th percentile) | 0 | 0 | - | NA |
| High in saturated fats (T0 cutoff: 99th percentile) | 0 | 0 | - | NA |
| High in sodium (T0 cutoff: 1st percentile) | 100 | 100 | - | 0 |

Values represent the frequency and 95% CI of "high in" products.

Cutoffs correspond to the limits on the amount of energy or nutrient of concern for the initial implementation of the law (i.e., for solids, per 100 g: 350 kcal of energy, 22.5 g of sugars, 6 g of saturated fats, 800 mg of sodium; for liquids, per 100 mL: 100 kcal of energy, 6 g of sugars, 3 g of saturated fats, 100 mg of sodium). The corresponding percentile was calculated according to T0 distribution of energy or nutrient of concern by food or beverage group.

Relative change: delta in the proportion between T0 and T1, relative to proportion in T0 (T0 − T1) × 100 ÷ T0; a negative sign represents a decrease, a positive sign represents an increase).

T0: preimplementation period, January to February 2015 + January to February 2016; T1: postimplementation period, January to February 2017 (*n* = 1,915).

Comparison between T0 and T1 were done using McNemar test.

Significant *p*-values (i.e., <0.05) are bolded.

Note that discrepancies in the percentage of "high in" sodium and any "high in" among soups were given by differences in the denominator used in each case; the denominator for the former did not consider food items with missing information of sodium, whereas the latter did.

on the food supply throughout the implementation of these phases will allow to test this hypothesis. Several bodies have also claimed that using cutoffs that are too strict (i.e., that leave most of a food category as regulated) would not incentivize food industry reformulation [35]; interestingly, we did not find that this was the case in Chile because we observed significant

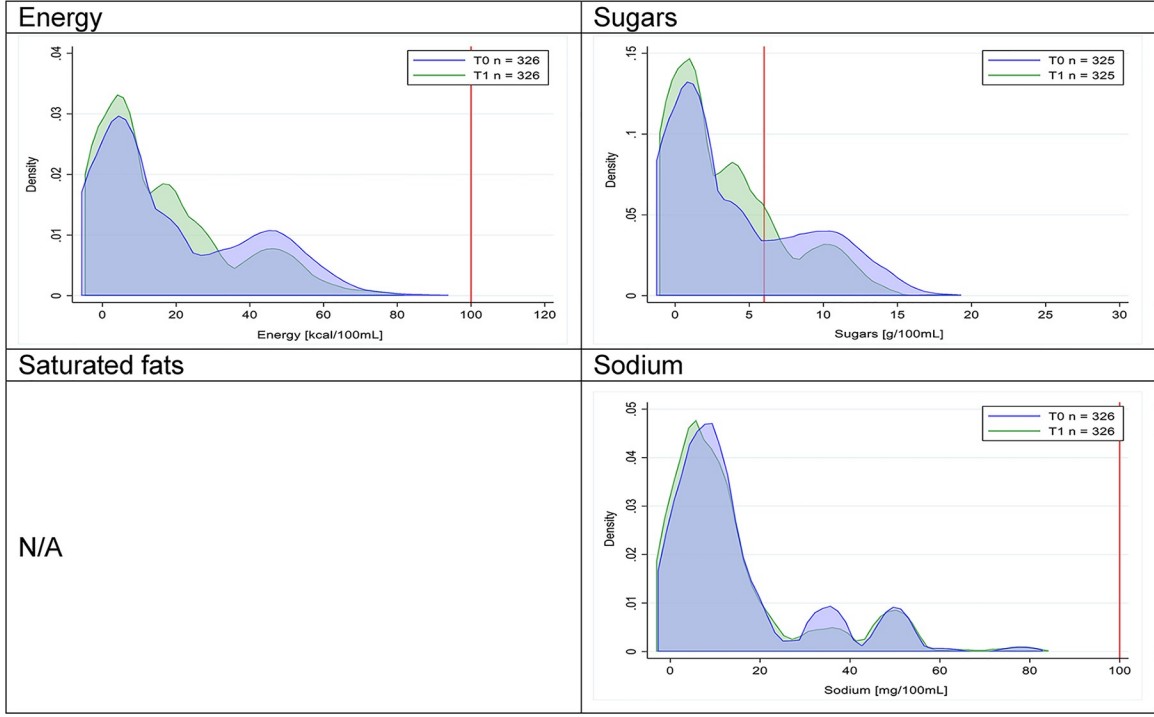

**Fig 18. Density curves for the amount of energy and nutrients of concern in beverages, longitudinal subsample.** The blue line represents the distribution in T0 (preimplementation), the green line represents the distribution in T1 (postimplementation), and the red line represents the cutoff for the amount of energy or nutrients of concern.

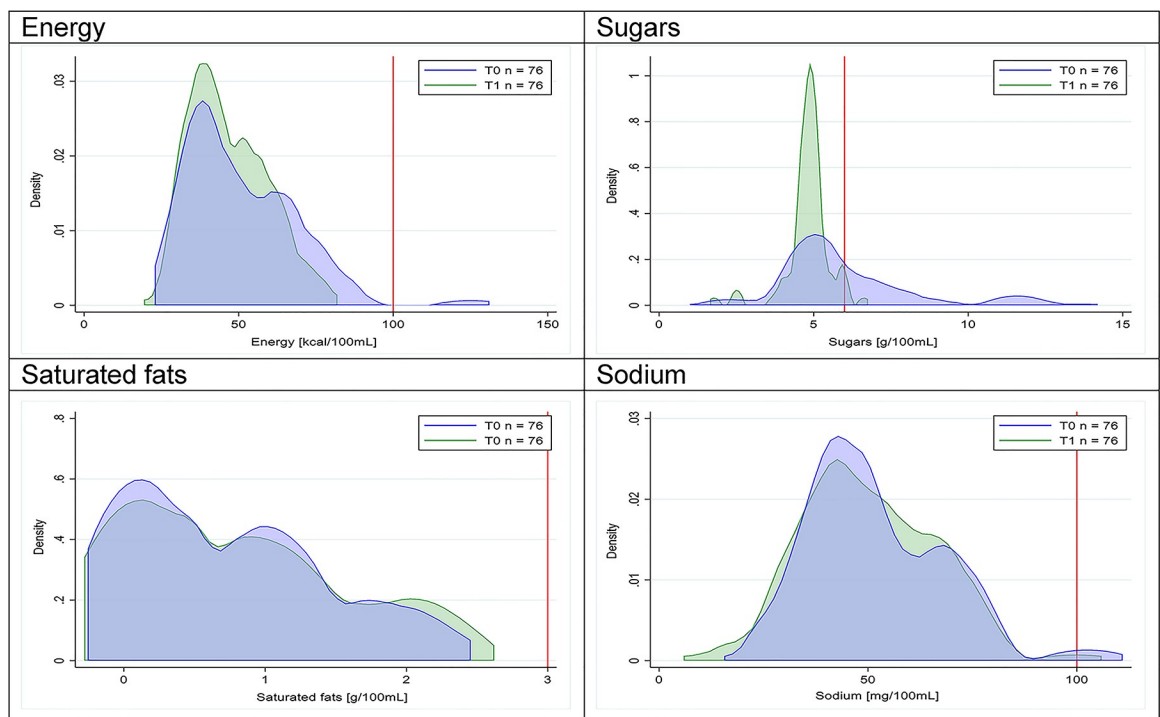

**Fig 19. . Density curves for the amount of energy and nutrients of concern in milk and milk-based drinks, longitudinal subsample.** The blue line represents the distribution in T0 (preimplementation), the green line represents the distribution in T1 (postimplementation), and the red line represents the cutoff for the amount of energy or nutrients of concern.

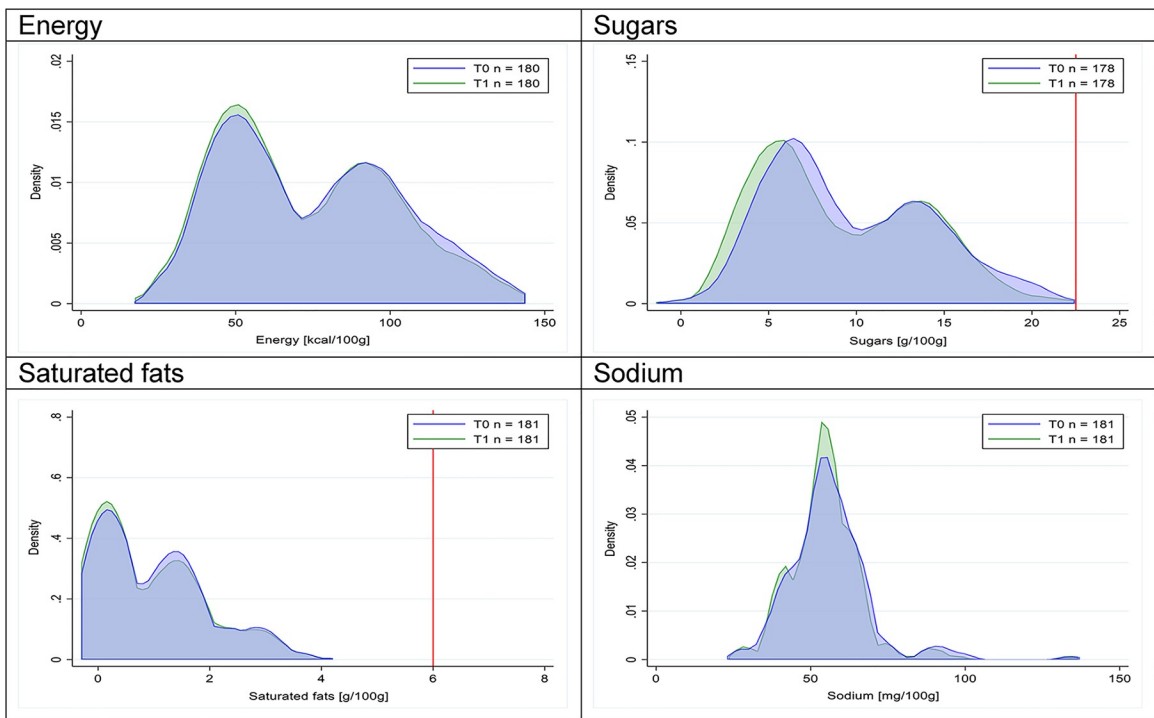

**Fig 20. Density curves for the amount of energy and nutrients of concern in yogurts, longitudinal subsample.** The blue line represents the distribution in T0 (preimplementation), the green line represents the distribution in T1 (postimplementation), and the red line represents the cutoff for the amount of energy or nutrients of concern.

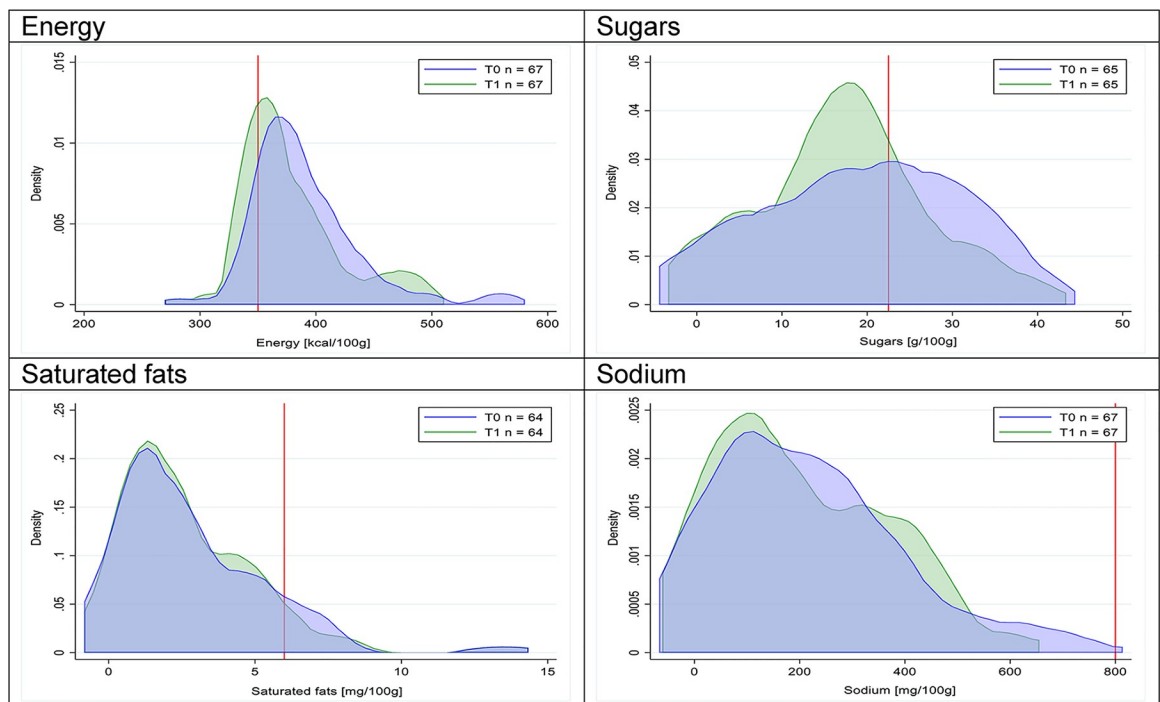

**Fig 21. Density curves for the amount of energy and nutrients of concern in breakfast cereals, longitudinal subsample.** The blue line represents the distribution in T0 (preimplementation), the green line represents the distribution in T1 (postimplementation), and the red line represents the cutoff for the amount of energy or nutrients of concern.

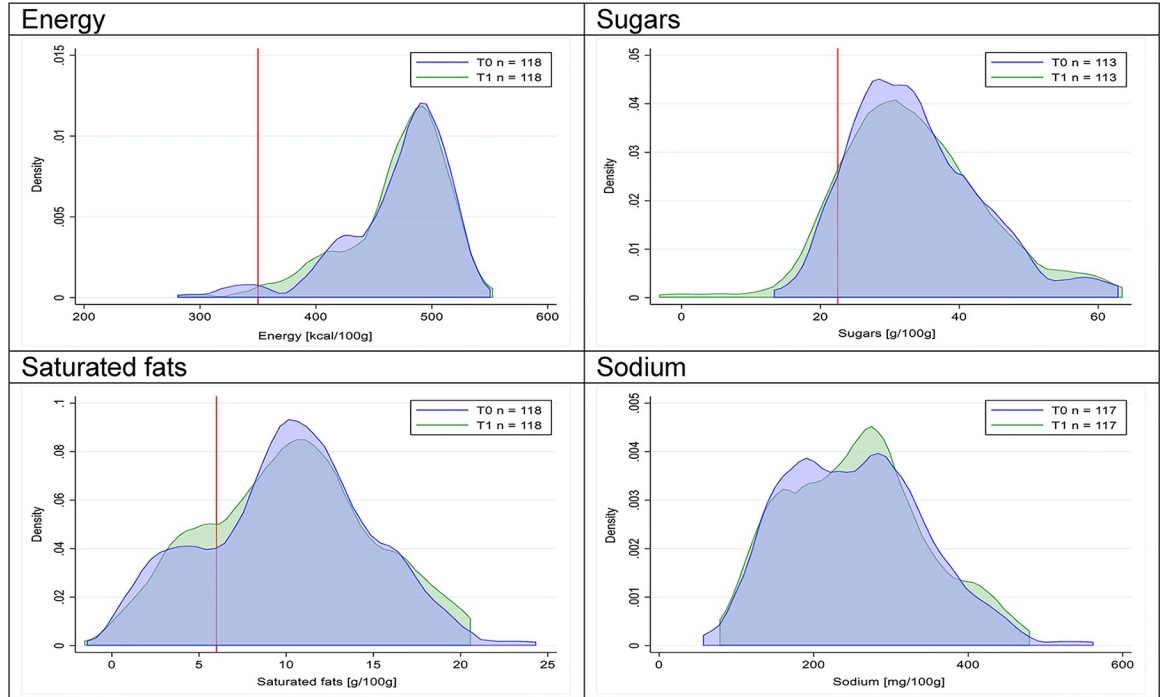

**Fig 22. Density curves for the amount of energy and nutrients of concern in sweet baked products, longitudinal subsample.** The blue line represents the distribution in T0 (preimplementation), the green line represents the distribution in T1 (postimplementation), and the red line represents the cutoff for the amount of energy or nutrients of concern.

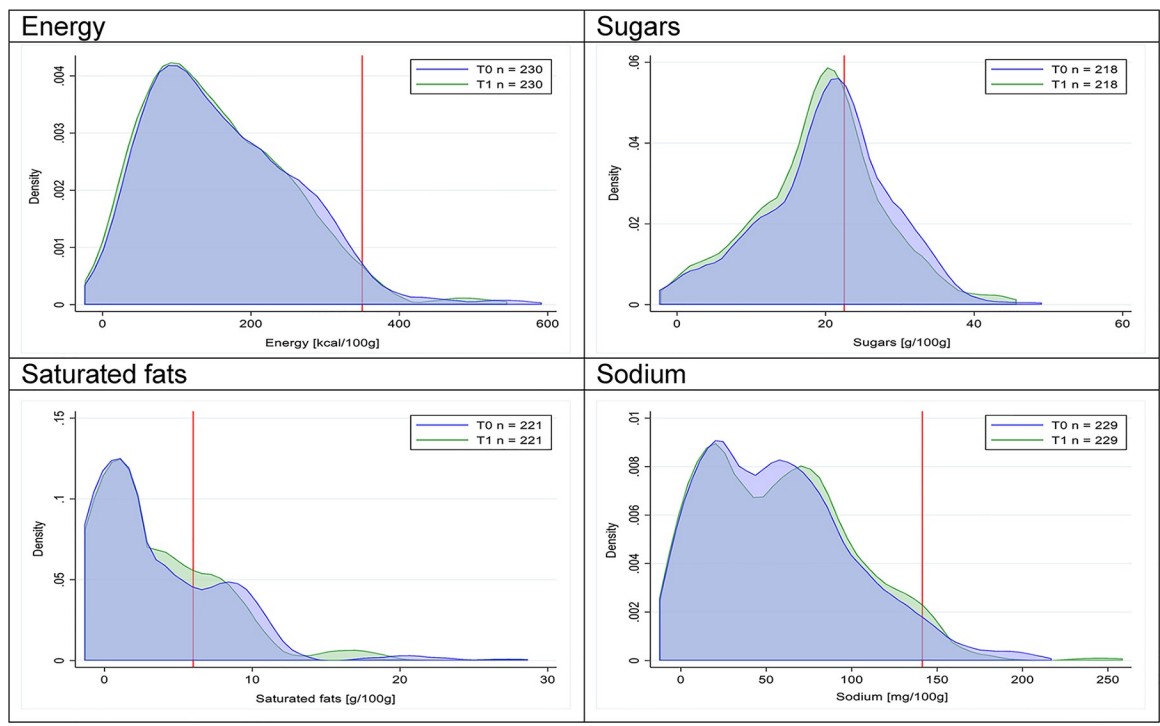

**Fig 23. Density curves for the amount of energy and nutrients of concern in desserts and ice creams, longitudinal subsample.** The blue line represents the distribution in T0 (preimplementation), the green line represents the distribution in T1 (postimplementation), and the red line represents the cutoff for the amount of energy or nutrients of concern.

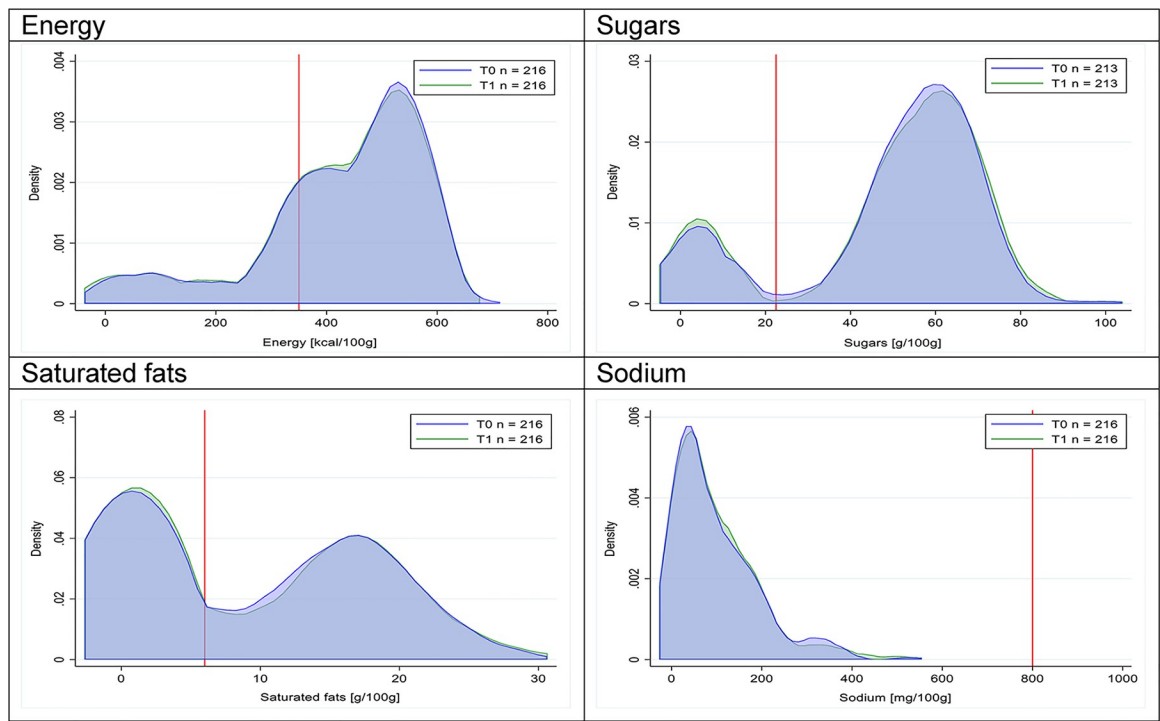

**Fig 24. Density curves for the amount of energy and nutrients of concern in candies and sweet confectioneries, longitudinal subsample.** The blue line represents the distribution in T0 (preimplementation), the green line represents the distribution in T1 (postimplementation), and the red line represents the cutoff for the amount of energy or nutrients of concern.

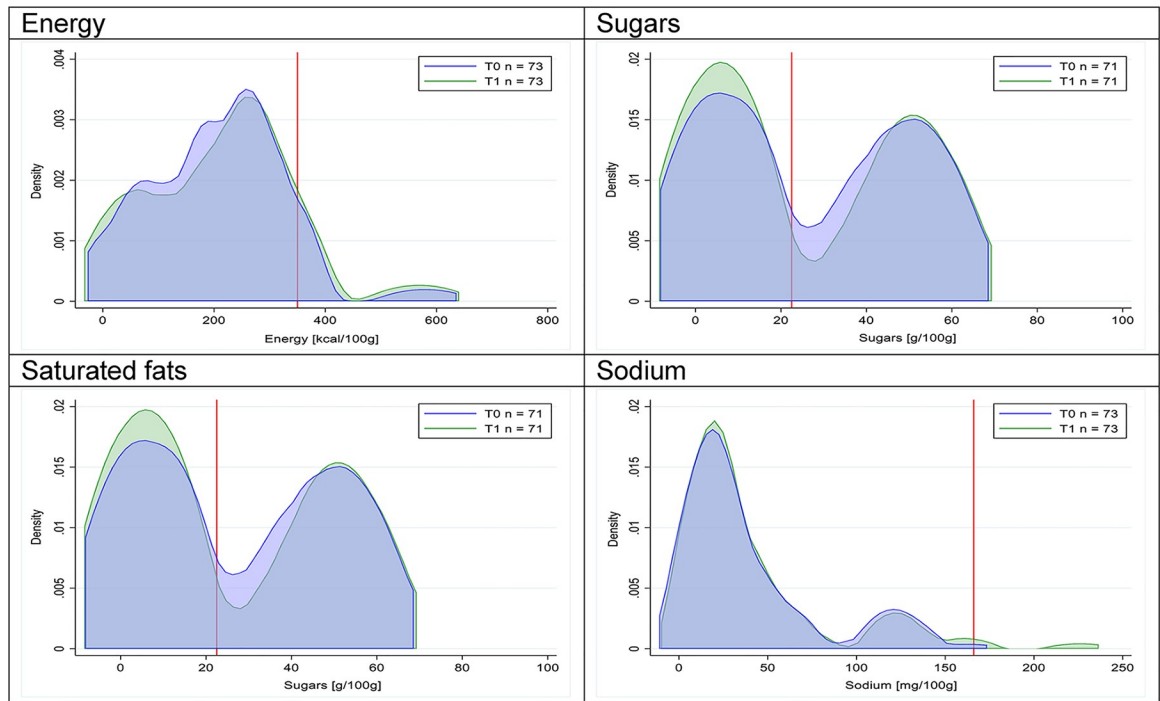

**Fig 25. Density curves for the amount of energy and nutrients of concern in sweet spreads, longitudinal subsample.** The blue line represents the distribution in T0 (preimplementation), the green line represents the distribution in T1 (postimplementation), and the red line represents the cutoff for the amount of energy or nutrients of concern.

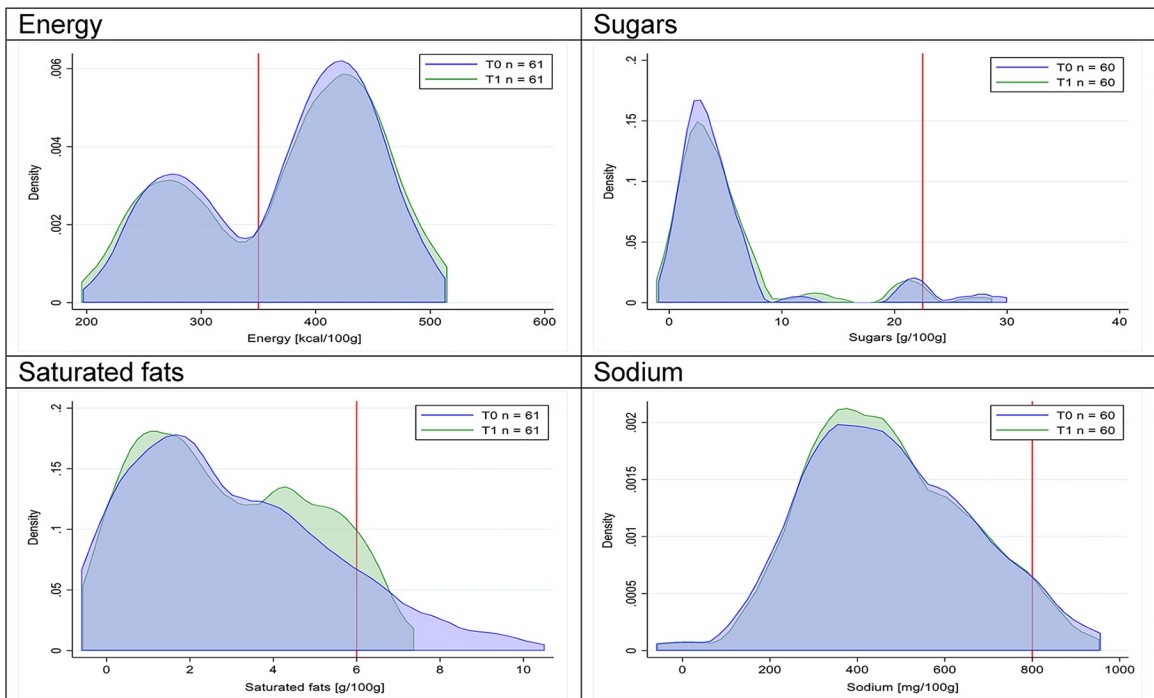

**Fig 26. Density curves for the amount of energy and nutrients of concern in savory baked products, longitudinal subsample.** The blue line represents the distribution in T0 (preimplementation), the green line represents the distribution in T1 (postimplementation), and the red line represents the cutoff for the amount of energy or nutrients of concern.

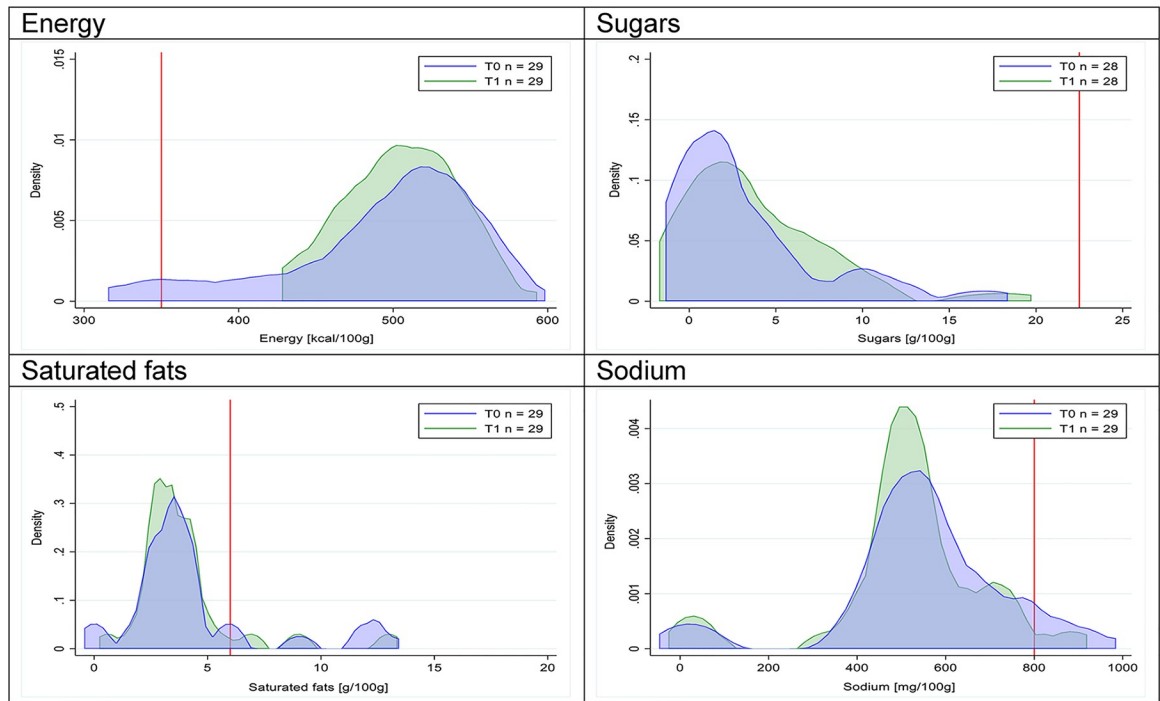

**Fig 27. Density curves for the amount of energy and nutrients of concern in savory snacks, longitudinal subsample.** The blue line represents the distribution in T0 (preimplementation), the green line represents the distribution in T1 (postimplementation), and the red line represents the cutoff for the amount of energy or nutrients of concern.

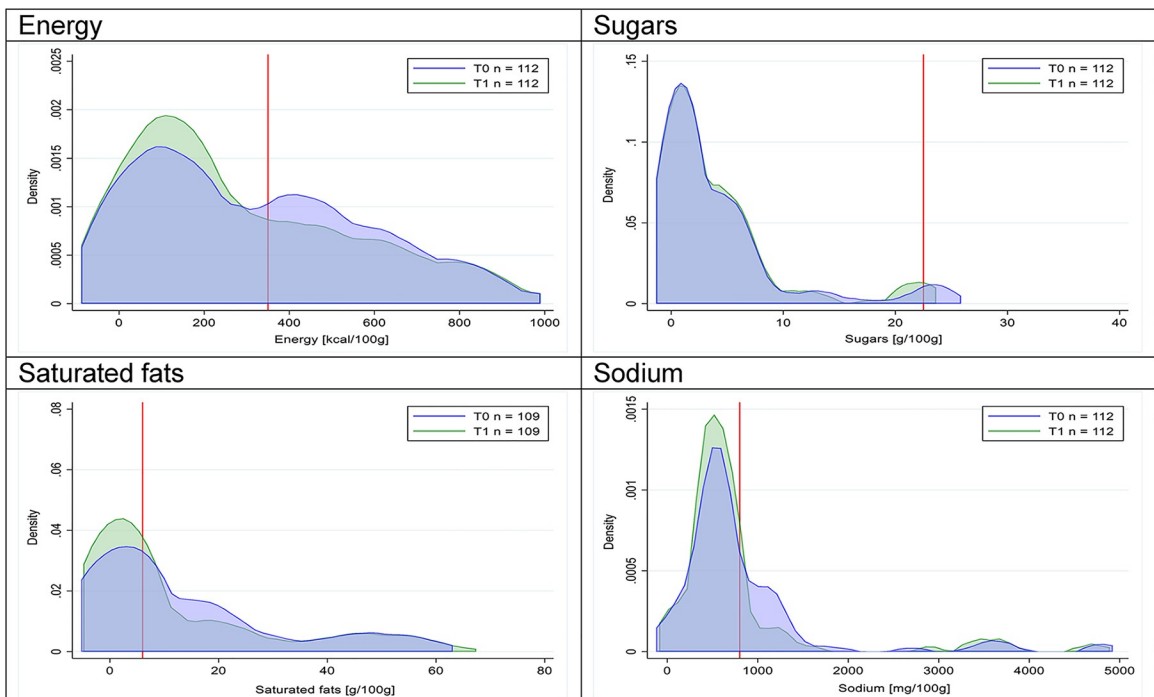

**Fig 28. Density curves for the amount of energy and nutrients of concern in savory spreads, longitudinal subsample.** The blue line represents the distribution in T0 (preimplementation), the green line represents the distribution in T1 (postimplementation), and the red line represents the cutoff for the amount of energy or nutrients of concern.

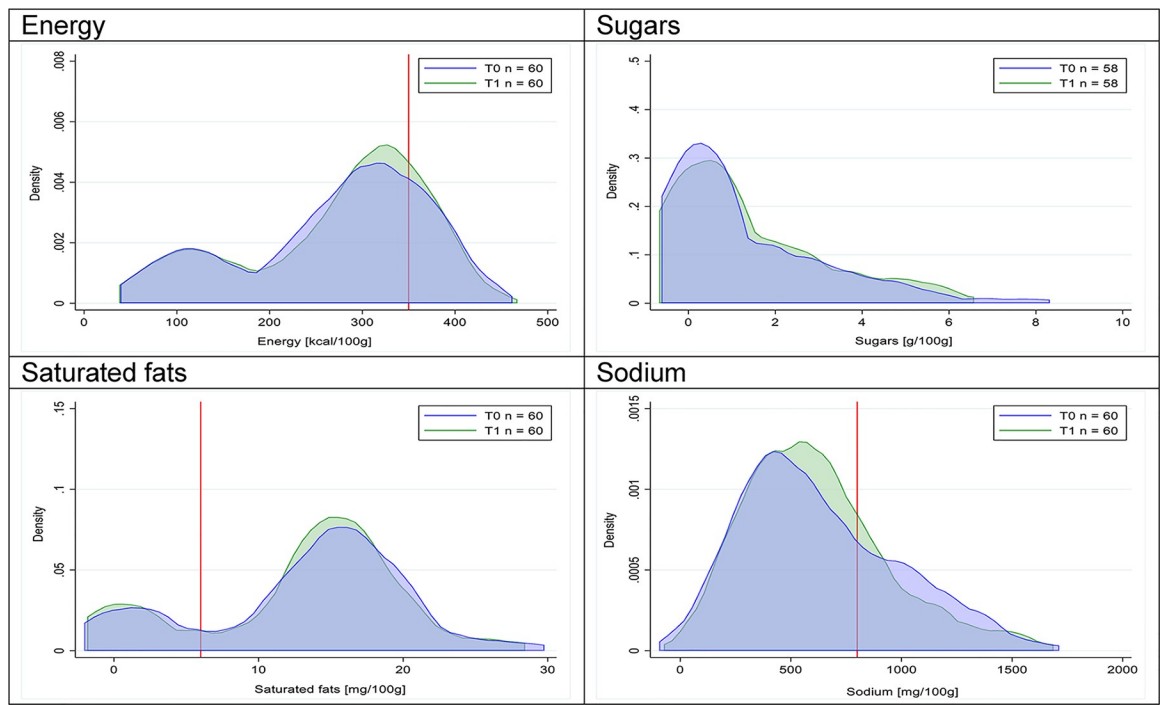

**Fig 29. Density curves for the amount of energy and nutrients of concern in cheeses, longitudinal subsample.** The blue line represents the distribution in T0 (preimplementation), the green line represents the distribution in T1 (postimplementation), and the red line represents the cutoff for the amount of energy or nutrients of concern.

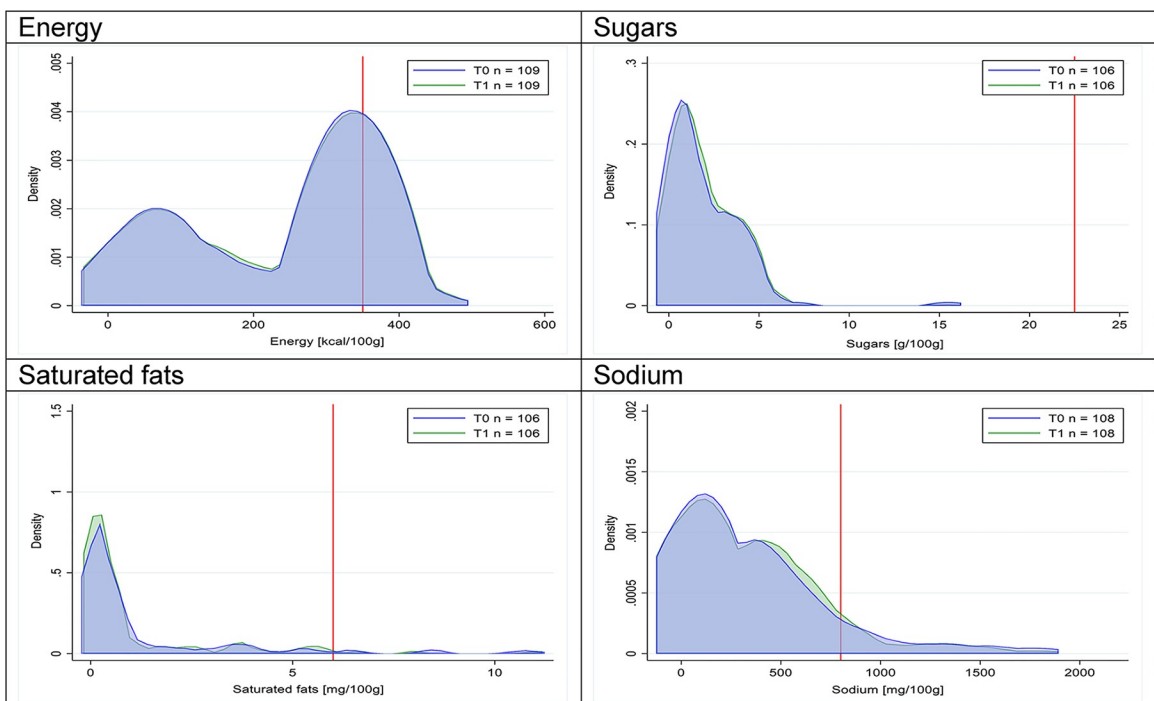

**Fig 30. Density curves for the amount of energy and nutrients of concern in ready-to-eat meals, longitudinal subsample.** The blue line represents the distribution in T0 (preimplementation), the green line represents the distribution in T1 (postimplementation), and the red line represents the cutoff for the amount of energy or nutrients of concern.

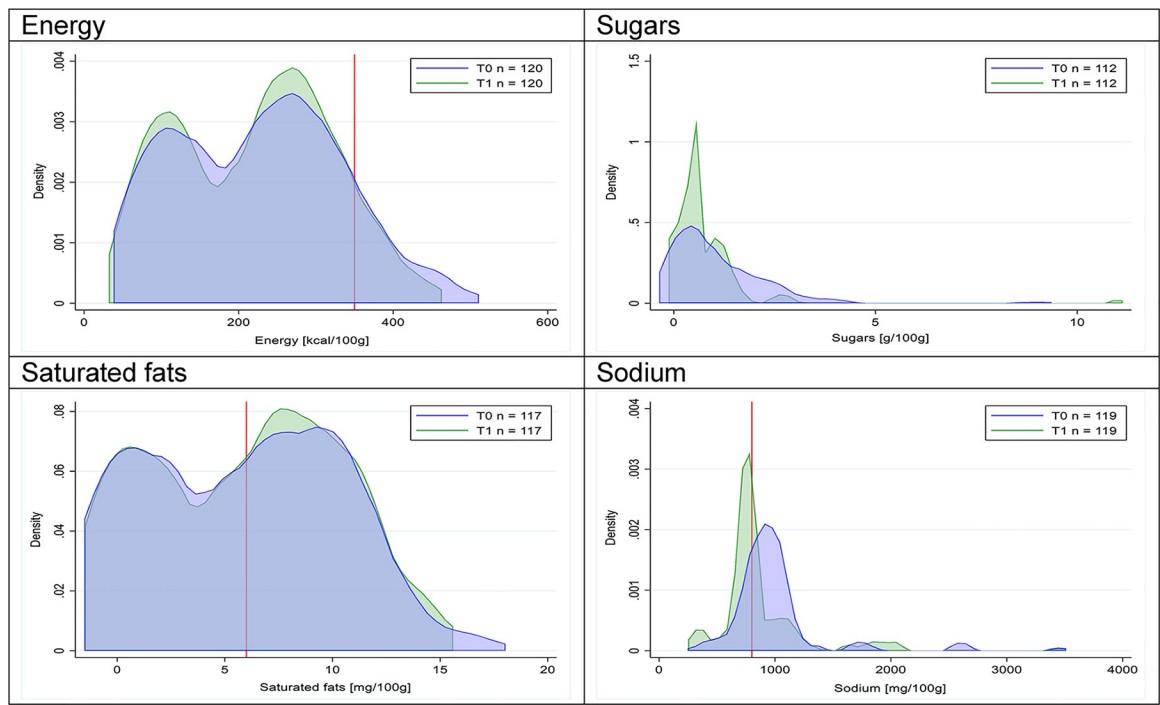

**Fig 31. Density curves for the amount of energy and nutrients of concern in sausages, longitudinal subsample.** The blue line represents the distribution in T0 (preimplementation), the green line represents the distribution in T1 (postimplementation), and the red line represents the cutoff for the amount of energy or nutrients of concern.

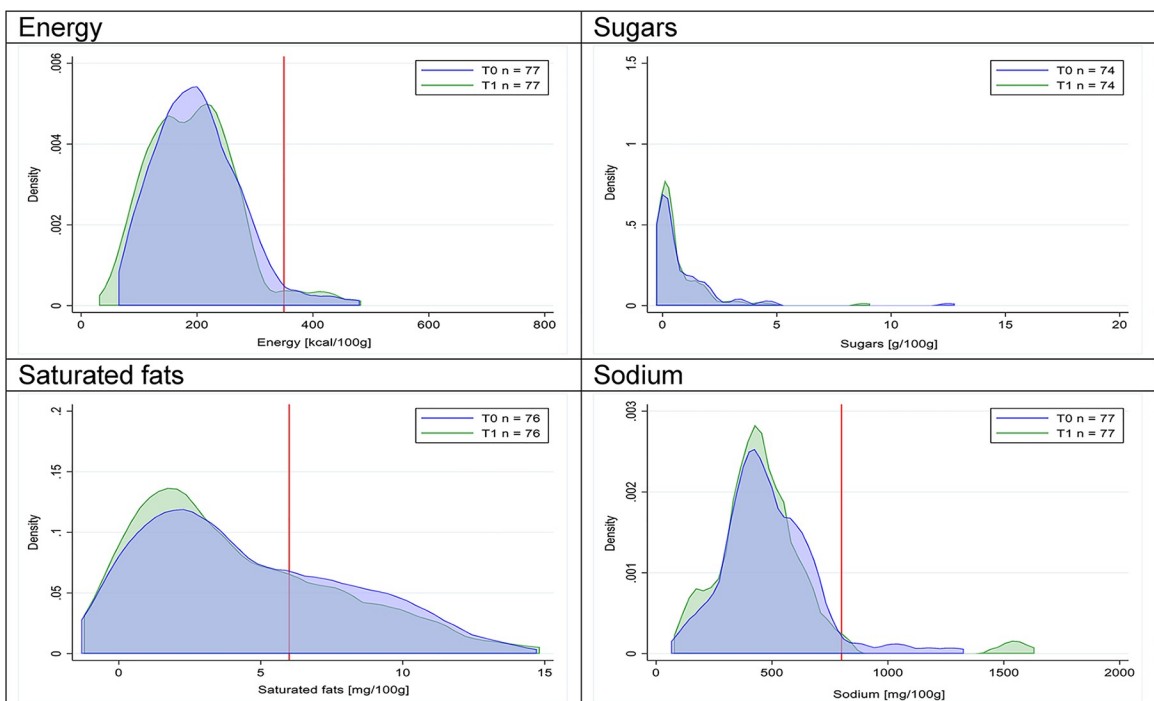

**Fig 32. Density curves for the amount of energy and nutrients of concern in nonsausage meat products, longitudinal subsample.** The blue line represents the distribution in T0 (preimplementation), the green line represents the distribution in T1 (postimplementation), and the red line represents the cutoff for the amount of energy or nutrients of concern.

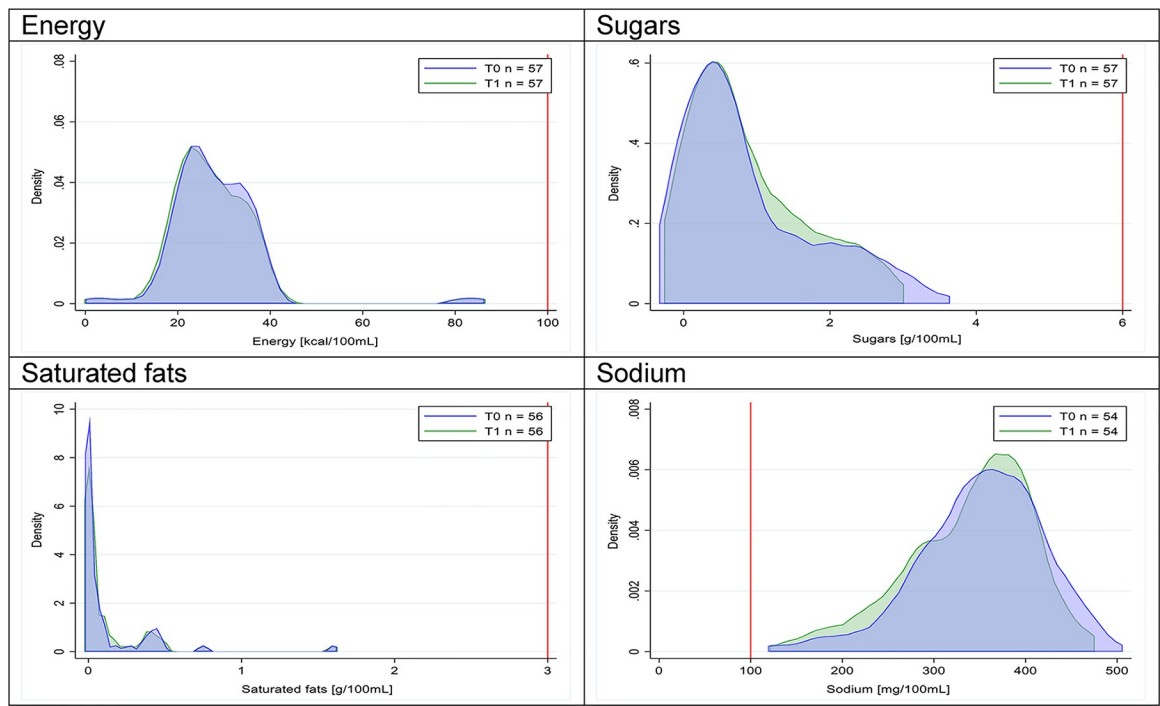

**Fig 33. Density curves for the amount of energy and nutrients of concern in soups, longitudinal subsample.** The blue line represents the distribution in T0 (preimplementation), the green line represents the distribution in T1 (postimplementation), and the red line represents the cutoff for the amount of energy or nutrients of concern.

changes in food categories in which cutoffs were below the 25th percentile (i.e., regulation affected 75% of the products of that category) such as sugars in sweet baked products, energy in breakfast cereals, and sodium in sausages and soups.

We believe our results suggest that industrial production of several packaged foods and beverages was affected by the initial implementation of the Chilean law. Although our design does not allow us to test causality, we do not believe the observed changes are due to previous trends of food reformulation because we have formerly shown that in the years prior to the implementation of the law there were no relevant changes on the amount of nutrients of concern of the food supply [28] and in the current analyses we do not observe significant changes in food categories in which cutoffs are at the top end of nutrient distribution. Moreover, the fact that most of the changes in the amount of energy or any nutrient of concern were around the first phase cutoffs is also suggestive of a promoting role of the regulatory process. The mandatory nature of the Chilean law and the fact that the regulation considered a set of diverse regulatory measures (i.e., FOP warning labels, marketing restrictions to children, and the healthy school food environment) may have resulted in this fast and significant reformulation of packaged food and beverage products. We believe our results show that food industry has the ability to reduce sugars and sodium amounts in major food and beverage groups, and therefore, efforts should be made to achieve these improvements as recently suggested [4]. At the same time, it is clear that the current efforts are limited in their effect on shifting the food supply from unhealthy saturated fats to healthier fats. In fact, the amount of saturated fats has increased in several food and beverage groups, which could reflect the technical challenges associated to replacement by, i.e., polyunsaturated fats that have a lower melting point and are less stable to oxidation, among other factors. Implication of this finding in population's health need to be elucidated and could depend on the source of the saturated fats involved [36].

**Table 4. Changes in quartiles of energy and nutrients of concern by food or beverages group, longitudinal analysis.**

| Beverages and foods | Energy (kcal/ 100 g-mL) | | | Sugars (g/ 100 g-mL) | | | Saturated fats (g/ 100 g-mL) | | | Sodium (mg/ 100 g-mL) | | |
|---|---|---|---|---|---|---|---|---|---|---|---|---|
| | T0 | T1 | *p*-value | T0 | T1 | *p*-value | T0 | T1 | *p*-value | T0 | T1 | *p*-value |
| **Beverages** | | | | | | | | | | | | |
| 25th percentile | 1.39 | 0.99 | 0.21 | 0.27 | 0.00 | 0.51 | - | - | - | 6.09 | 6.00 | 0.88 |
| 50th percentile | 15.65 | 14.80 | 0.22 | 1.34 | 0.01 | 0.06 | - | - | - | 8.37 | 8.00 | 0.39 |
| 75th percentile | 24.85 | 23.80 | 0.34 | **6.03** | **5.07** | **0.02** | - | - | - | 20.10 | 19.85 | 0.54 |
| **Milks and milk-based drinks** | | | | | | | | | | | | |
| 25th percentile | 41.92 | 41.29 | 0.63 | **5.00** | **4.40** | **<0.01** | 1.00 | 0.93 | 0.21 | 46.15 | 45.85 | 0.79 |
| 50th percentile | 47.00 | 44.81 | 0.21 | **6.00** | **5.00** | **0.01** | 1.00 | 0.94 | 0.34 | 49.98 | 49.20 | 0.48 |
| 75th percentile | **55.47** | **50.51** | **0.04** | **6.99** | **5.55** | **<0.01** | 1.04 | 1.00 | 0.51 | 56.60 | 55.89 | 0.60 |
| **Yogurts** | | | | | | | | | | | | |
| 25th percentile | **75.43** | **69.43** | **0.03** | 6.99 | 6.95 | 0.92 | 0.99 | 0.95 | 0.21 | 52.87 | 52.28 | 0.58 |
| 50th percentile | 75.81 | 73.29 | 0.40 | 9.00 | 8.81 | 0.43 | 1.00 | 0.96 | 0.14 | 56.05 | 55.54 | 0.31 |
| 75th percentile | 90.00 | 89.53 | 0.76 | 11.97 | 11.10 | 0.03 | 1.03 | 0.98 | 0.25 | 56.14 | 55.00 | 0.35 |
| **Breakfast cereals** | | | | | | | | | | | | |
| 25th percentile | **358.99** | **348.99** | **0.03** | 13.38 | 12.99 | 0.75 | 2.00 | 2.00 | 1.00 | 153.54 | 141.46 | 0.47 |
| 50th percentile | **388.16** | **376.81** | **0.04** | 17.00 | 15.78 | 0.12 | 3.00 | 2.85 | 0.33 | 214.12 | 200.80 | 0.36 |
| 75th percentile | **411.51** | **393.61** | **<0.01** | 22.00 | 19.32 | 0.06 | **4.02** | **3.52** | **0.02** | 284.82 | 267.43 | 0.31 |
| **Sweet baked products** | | | | | | | | | | | | |
| 25th percentile | 454.65 | 454.50 | 0.93 | 29.95 | 29.71 | 0.49 | 7.00 | 6.99 | 1.00 | 240.00 | 237.25 | 0.26 |
| 50th percentile | 473.51 | 472.49 | 0.36 | 34.01 | 33.95 | 0.83 | 10.11 | 10.00 | 0.48 | 248.00 | 247.29 | 0.94 |
| 75th percentile | 481.99 | 480.53 | 0.33 | 35.50 | 35.29 | 0.43 | 11.50 | 11.46 | 0.80 | 264.92 | 263.92 | 0.58 |
| **Dessert and ice creams** | | | | | | | | | | | | |
| 25th percentile | 129.76 | 128.00 | 0.44 | 19.12 | 18.80 | 0.44 | 0.02 | 0.00 | 0.90 | 45.53 | 44.02 | 0.14 |
| 50th percentile | 160.18 | 153.78 | 0.07 | 19.99 | 19.41 | 0.09 | 3.60 | 3.54 | 0.79 | 52.96 | 52.78 | 0.86 |
| 75th percentile | 185.89 | 175.00 | 0.22 | 22.31 | 22.00 | 0.46 | **10.36** | **8.69** | **0.02** | 73.99 | 73.00 | 0.33 |
| **Candies and sweet confectionaries** | | | | | | | | | | | | |
| 25th percentile | 347.55 | 344.77 | 0.39 | 32.56 | 31.92 | 0.26 | 6.12 | 6.05 | 0.92 | 90.50 | 89.59 | 0.66 |
| 50th percentile | **431.24** | **425.38** | **0.04** | 47.75 | 47.61 | 0.73 | 8.50 | 8.50 | 1.00 | 97.66 | 97.44 | 0.82 |
| 75th percentile | 450.46 | 449.54 | 0.72 | 50.14 | 49.72 | 0.44 | 11.31 | 10.74 | 0.36 | 117.74 | 114.77 | 0.07 |
| **Sweet spreads** | | | | | | | | | | | | |
| 25th percentile | 118.03 | 116.59 | 0.88 | 3.92 | 3.26 | 0.37 | 0.00 | 0.00 | 1.00 | 14.53 | 14.00 | 0.60 |
| 50th percentile | 225.03 | 221.36 | 0.47 | 27.45 | 27.44 | 0.99 | 7.05 | 6.14 | 0.26 | 30.99 | 30.81 | 0.79 |
| 75th percentile | 239.92 | 237.30 | 0.60 | 54.82 | 54.28 | 0.75 | 11.98 | 10.52 | 0.49 | 50.40 | 50.01 | 0.66 |
| **Savory baked products** | | | | | | | | | | | | |
| 25th percentile | 337.07 | 334.79 | 0.34 | 2.00 | 2.00 | 1.00 | 1.00 | 0.99 | 1.00 | 392.86 | 386.16 | 0.07 |
| 50th percentile | 406.00 | 404.52 | 0.60 | 3.00 | 3.00 | 0.99 | 2.90 | 3.00 | 1.00 | 448.00 | 445.07 | 0.62 |
| 75th percentile | 416.12 | 414.05 | 0.28 | 8.12 | 7.96 | 0.62 | 3.75 | 3.44 | 0.17 | 571.01 | 565.99 | 0.30 |
| **Savory snacks** | | | | | | | | | | | | |
| 25th percentile | 481.98 | 472.49 | 0.36 | 0.14 | 0.72 | 0.12 | 2.99 | 2.99 | 1.00 | 485.00 | 484.99 | 1.00 |
| 50th percentile | 507.02 | 504.00 | 0.50 | **2.62** | **3.60** | **<0.01** | 3.79 | 3.50 | 0.44 | 538.16 | 509.99 | 0.17 |
| 75th percentile | 513.87 | 513.84 | 0.99 | 4.37 | 5.00 | 0.07 | 5.18 | 5.18 | 1.00 | 636.17 | 609.39 | 0.24 |
| **Savory spreads** | | | | | | | | | | | | |
| 25th percentile | 69.00 | 67.66 | 0.84 | 0.09 | 0.00 | 0.54 | **1.47** | **0.00** | **0.03** | **471.35** | **399.00** | **0.01** |
| 50th percentile | 298.93 | 283.60 | 0.07 | 3.00 | 3.00 | 1.00 | 11.20 | 11.20 | 1.00 | 596.52 | 576.44 | 0.56 |
| 75th percentile | 422.16 | 415.05 | 0.47 | 6.33 | 6.33 | 1.00 | **13.10** | **12.90** | **0.04** | 933.19 | 744.99 | 0.06 |
| **Cheeses** | | | | | | | | | | | | |
| 25th percentile | 223.52 | 222.49 | 0.75 | 0.03 | 0.00 | 0.81 | 12.05 | 11.71 | 0.55 | 484.96 | 483.96 | 0.97 |

(*Continued*)

**Table 4.** (Continued)

| Beverages and foods | Energy (kcal/ 100 g-mL) | | | Sugars (g/ 100 g-mL) | | | Saturated fats (g/ 100 g-mL) | | | Sodium (mg/ 100 g-mL) | | |
|---|---|---|---|---|---|---|---|---|---|---|---|---|
| | T0 | T1 | p-value | T0 | T1 | p-value | T0 | T1 | p-value | T0 | T1 | p-value |
| 50th percentile | 269.14 | 266.00 | 0.48 | 1.34 | 1.27 | 0.66 | 13.99 | 13.98 | 0.91 | 596.61 | 596.52 | 0.98 |
| 75th percentile | 338.00 | 336.98 | 0.79 | 1.95 | 1.65 | 0.12 | 17.02 | 17.00 | 0.97 | 794.00 | 770.65 | 0.62 |
| **Ready-to-eat meals** | | | | | | | | | | | | |
| 25th percentile | 191.05 | 177.95 | 1.00 | 1.27 | 1.00 | 0.60 | 0.01 | 0.00 | 0.28 | 107.88 | 5.00 | 0.80 |
| 50th percentile | 243.35 | 234.38 | 0.81 | 2.01 | 1.98 | 0.80 | 1.00 | 0.98 | 0.86 | 281.92 | 259.11 | 0.65 |
| 75th percentile | 342.45 | 337.99 | 0.10 | 2.50 | 2.50 | 1.00 | 1.34 | 1.16 | 0.52 | 451.70 | 441.94 | 0.84 |
| **Sausages** | | | | | | | | | | | | |
| 25th percentile | 195.81 | 193.77 | 0.50 | 0.99 | 0.99 | 1.00 | 4.88 | 4.40 | 0.05 | **817.26** | **713.00** | **<0.01** |
| 50th percentile | 219.21 | 218.80 | 0.89 | 1.00 | 1.00 | 0.98 | 5.50 | 5.35 | 0.51 | **931.28** | **781.98** | **<0.01** |
| 75th percentile | 262.65 | 257.64 | 0.16 | 1.53 | 1.34 | 0.14 | 9.07 | 8.97 | 0.70 | 1,041.19 | 927.98 | 0.05 |
| **Nonsausage meat products** | | | | | | | | | | | | |
| 25th percentile | 176.61 | 168.43 | 0.05 | 0.00 | 0.00 | 1.00 | 2.00 | 2.00 | 0.99 | **390.00** | **360.00** | **0.03** |
| 50th percentile | 189.56 | 181.50 | 0.07 | 0.00 | 0.00 | 1.00 | 4.66 | 4.55 | 0.99 | 466.96 | 462.00 | 0.65 |
| 75th percentile | 248.17 | 247.02 | 0.82 | 1.00 | 1.00 | 1.00 | **7.16** | **6.33** | **0.02** | 491.10 | 480.00 | 0.44 |
| **Soups** | | | | | | | | | | | | |
| 25th percentile | 23.04 | 22.97 | 0.86 | 0.99 | 0.97 | 0.76 | 0.00 | 0.00 | 0.98 | 318.99 | 312.54 | 0.16 |
| 50th percentile | **27.00** | **25.99** | **0.04** | 0.99 | 0.98 | 0.77 | 0.00 | 0.00 | 0.20 | 372.11 | 368.90 | 0.48 |
| 75th percentile | 32.02 | 32.01 | 0.98 | 1.19 | 0.99 | 0.21 | 0.01 | 0.00 | 0.13 | 377.73 | 374.68 | 0.50 |

Quartiles and p-values were obtained from quantile regression for the linear mixed-effect models (one model per nutrient per food or beverage group), using implementation period as independent variable (T0 = 0, T1 = 1).

Significant p-values (i.e., <0.05) are bolded.

T0: preimplementation period, January to February 2015 + January to February 2016 (n = 1,915).

T1: postimplementation period, January to February 2017 (n = 1,915).

These results can be useful to other countries that are on their way to implementing or have already implemented the use of warning labels for unhealthy packaged foods to prevent obesity, such as Peru, Uruguay, Israel, Mexico, Canada, and Brazil. A comparative study of the healthiness of packaged foods and beverages from 12 countries reported a poor HSR score for the packaged food supply in Chile in 2015 (only Hong Kong and India had poorer scores) [37], suggesting the external validity of our findings could vary from country to country.

Recent reports indicate that food reformulation has a profound impact on diet quality; although this may be an area of debate [4, 8]. In the case of Chile, according to the National Dietary Survey (2010), beverages, milks and milk-based drinks, breakfast cereals, sweet baked products, and sweet spreads represent approximately 15% of the calories and approximately 40% of the added sugars consumed daily by the Chilean population, whereas savory spreads and sausages account for 22% and 8% of sodium intake, respectively [38, 39]. Therefore, the observed decrease in median amount of energy and sugars have the potential to help mitigate the excessive dietary intakes in added sugars [38]. Moreover, in a sample of Chilean preschoolers, it has been described that desserts, dairy products, and beverages cover approximately 23% to 30% of the intake of calories [40], therefore suggesting that the impact could even be higher among children. In the case of sodium, no information is available on the share of the sodium intake represented by the food groups in the study in the Chilean diet; thus, we cannot estimate the potential dietary impact of these changes on sodium intake.

There are limitations in this study. We cannot truly show a direct linkage between the initial implementation of the law and the changes in the amount of energy and nutrients of concern

of packaged foods; however, the fact that most left shifts of the distributions were around the first phase cutoffs suggest the regulation might be a relevant driver. Also, we cannot disentangle the effect on reformulation of the different policies implemented simultaneously. Other limitations are that our results are not sales-weighted nor consider the dietary share in Chilean diets. However, our analytical sample included only foods and beverages representing ≥1% of the sales of the specific category in order to obtain more meaningful results [25]. We primarily conducted cross-sectional analyses in which best-selling products were compared before and after the implementation of the law. However, by taking this approach, we were unable to differentiate whether changes are due to reformulation of existing products or to exit of old products or entry of new products; moreover, we are unable to fully exclude sampling differences between years. Therefore, we also conducted longitudinal analyses including only food and beverage products that were available before and after the implementation of the law, which allows us to assess reformulation. We believe consistency between cross-sectional and longitudinal analyses increases the validity of our findings. Also, our analysis did not include confounders or modifiers such as manufacturer company. Finally, our analyses rely on the nutrient amount reported by the food manufacturer on the nutrition facts panel printed on the package and is not based on laboratory assessment. A major strength of our study, however, is that all study data were collected prospectively versus one that could be obtained retrospectively from food industry or retailers' databases, therefore decreasing its error.

In conclusion, our results show that several food and beverage groups available in the Chilean market decreased the proportion of products "high in" after a short-time period (<1 year) of implementing a mandatory set of regulations including simple FOP warning labels for energy, sodium, total sugars, and saturated fats. These changes happened mainly for sugars and sodium and were reflected in significant decrease in the amount of such nutrients close to the initial cutoff for defining unhealthy foods and beverages. It remains to be seen how consumers react to these food composition changes and whether this short-term reformulation is sustained over time, especially with the complete implementation of the law. Also, future studies should investigate whether the reported reformulation either positively or negatively impacted the quality of the food supply more broadly, considering, i.e., other nutrients or food components, such as nonnutritive sweeteners. More importantly, future studies need to focus on whether the improvements in sugars or sodium observed among foods and beverages ultimately impacts the quality of the overall Chilean diet.

## Supporting information

**S1 Checklist. STROBE statement: Checklist of items that should be included in reports of cross-sectional studies.**
(DOC)

**S2 Checklist. STROBE statement: Checklist of items that should be included in reports of cohort studies.**
(DOC)

**S1 Table. Cutoffs for defining products "high in" in every implementation phase.**
(DOCX)

**S2 Table. Type of foods classified in each food or beverage group.** T0: preimplementation period, January to February 2015 + January to February 2016 (cross-sectional $n$ = 4,055, longitudinal $n$ = 1,915). T1: postimplementation period, January to February 2017 (cross-sectional $n$ = 3,025, longitudinal $n$ = 1,915).
(DOCX)

**S3 Table. Outliers and implausible values for the amount of energy and nutrients of concern.**
(DOCX)

**S1 Text. Concept note including first draft of data analysis.**
(DOCX)

## Acknowledgments

We thank the Chilean National Association for Supermarkets (ASACH) and all the supermarkets and candy distributors involved for authorizing the data collection. We also thank the research teams at CIAPEC (Center of Research in Food Environment and Prevention of Obesity and Non-Communicable Diseases) at INTA (Institute of Nutrition and Food Technology), University of Chile, and at the Global Food Research Program, University of North Carolina at Chapel Hill.

## Author Contributions

**Conceptualization:** Marcela Reyes, Lindsey Smith Taillie, Barry Popkin, Rebecca Kanter, Stefanie Vandevijvere, Camila Corvalán.

**Data curation:** Marcela Reyes, Rebecca Kanter.

**Formal analysis:** Marcela Reyes, Lindsey Smith Taillie, Rebecca Kanter, Camila Corvalán.

**Funding acquisition:** Marcela Reyes, Lindsey Smith Taillie, Barry Popkin, Rebecca Kanter, Stefanie Vandevijvere, Camila Corvalán.

**Investigation:** Marcela Reyes, Lindsey Smith Taillie, Barry Popkin, Rebecca Kanter, Stefanie Vandevijvere, Camila Corvalán.

**Methodology:** Marcela Reyes, Lindsey Smith Taillie, Barry Popkin, Rebecca Kanter, Stefanie Vandevijvere, Camila Corvalán.

**Project administration:** Marcela Reyes, Lindsey Smith Taillie, Camila Corvalán.

**Supervision:** Rebecca Kanter.

**Writing – original draft:** Marcela Reyes, Barry Popkin, Camila Corvalán.

**Writing – review & editing:** Marcela Reyes, Lindsey Smith Taillie, Barry Popkin, Rebecca Kanter, Stefanie Vandevijvere, Camila Corvalán.

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
