## [Editor Report · Decision Letter 0]

7 Feb 2020

Dear Dr Corvalan, 

Thank you for submitting your manuscript entitled "Changes on the nutrient content of packaged foods and beverages after the initial implementation of the Chilean Law of Food Labelling and Advertising: a non-experimental prospective study." for consideration by PLOS Medicine.

Your manuscript has now been evaluated by the PLOS Medicine editorial staff [as well as by an academic editor with relevant expertise] and I am writing to let you know that we would like to send your submission out for external peer review.

Kind regards,

Adya Misra, PhD,

Senior Editor

PLOS Medicine

---

## [Decision Letter · Decision Letter 1]

30 Apr 2020

Dear Dr. Corvalan,

Thank you very much for submitting your manuscript "Changes on the nutrient content of packaged foods and beverages after the initial implementation of the Chilean Law of Food Labelling and Advertising: a non-experimental prospective study." (PMEDICINE-D-20-00319R1) for consideration at PLOS Medicine. 

Your paper was evaluated by a senior editor and discussed among all the editors here. It was also discussed with an academic editor with relevant expertise, and sent to independent reviewers, including a statistical reviewer. We feel that your study would be well suited to our special issue focussing on obesity which is due to publish in July 2020. Please let me know if you would be interested in being included in this issue. The reviews are appended at the bottom of this email and any accompanying reviewer attachments can be seen via the link below:

[LINK]

In light of these reviews, I am afraid that we will not be able to accept the manuscript for publication in the journal in its current form, but we would like to consider a revised version that addresses the reviewers' and editors' comments. Obviously we cannot make any decision about publication until we have seen the revised manuscript and your response, and we plan to seek re-review by one or more of the reviewers. 

We expect to receive your revised manuscript by May 21 2020 11:59PM. Please email us (plosmedicine@plos.org) if you have any questions or concerns.

We look forward to receiving your revised manuscript. 

Sincerely,

Adya Misra, PhD

Senior Editor 

PLOS Medicine

plosmedicine.org

Abstract

Please combine methods and findings section to adhere to PLOS Medicine style

Please mention the data sources used 

The last sentence of the methods and findings section must include a limitation of your study design/methodology 

Please provide 95% CI and p -values as needed when reporting numerical results or results of “significance”

Please replace the interpretation section with conclusions and begin this section with “our results show” or similar to avoid overreaching 

Please remove funding information and provide this information in the funding section within the metadata

Data Availability

The Data Availability Statement (DAS) requires revision. For each data source used in your study: 

On page 3 please replace the summary sub heading with “Abstract”

Author summary

Introduction

Line 53 please revise to “In the last few decades …” or similar 

Please format all references according to Vancouver style and use square brackets within text

Lines 81-82 could you mention the years studied here to avoid ambiguity 

Methods

Please mention names of all data sources used 

Role of the funding source should be moved to the funding disclosure section in the article metadata

Results 

Please ensure all numerical results are accompanied by p values and 95% CI when noting “significance” or the lack of it. For example Lines 200-201 

Please provide exact p values, unless p<0.001 

Discussion

Please avoid assertions of primacy (Line 243) and add ” to our knowledge..”

I note that potential confounders and effect modifiers were not assessed- should this be added as a limitation to the discussion?

Please ensure that the study is reported according to the STROBE guideline, and include the completed [STROBE or other] checklist as Supporting Information. When completing the checklist, please use section and paragraph numbers, rather than page numbers. Please add the following statement, or similar, to the Methods: "This study is reported as per the Strengthening the Reporting of Observational Studies in Epidemiology STROBE guideline (S1 Checklist)."

Please report your study according to the relevant guideline, which can be found here: http://www.equator-network.org/

Comments from the reviewers:

Reviewer #1: This paper studies the changes in the proportion of food/beverages exceeding the cut-off of the Chilean Food Labelling and Advertising Law as well as the changes median content of energy and nutrients of these products after the implementation of this Law. It offers important contribution to literature as it provides the highly needed evidence on the potential effectiveness of labelling regulations on improving the nutritional quality of packaged foods and beverages. Attention to the following would improve the manuscript: 

1. A large part of the analysis mentions the relative percentage changes of the outcomes between the two periods. However, related figures are not given in tables 1 and 2, making it difficult for readers to follow. The analysis will therefore be clearer if additional columns are added for both samples to provide these percentage change figures. 

2. One key finding of the paper is that the content of nutrients of most food groups decreased just below the cut-off from the visual analysis. However, this finding is not clearly explained in the paper, such as how to determine if the shift is just below the cut-off. Given the large numbers of graphs presented, it would also be helpful to have some indications on which graphs display such left-shifts. 

3. Some food groups displayed a significant increase in saturated fats and energy after the implemented the law. The authors should provide some discussions on the potential reasons for these increases and their implications on the overall nutritional quality of packaged foods or beverages. 

4. Line 131 - where is the sales data sourced from?

5. The titles for figures 1 and 2 are missing.

6. Line 224 - is the "5% change" an increase or a decrease?

7. Line 247 - which nutrient is "the proportion of 'high-in' products" refereed to? 

Reviewer #2: I confine my remarks to statistical aspects of this paper. Unfortunately, I think that the wrong variable was analyzed.

When I first read the description of the project, I thought that, if I ran a food company, I would just lower all my numbers to just below the cutoffs. Then the product would not be marked as "high" in anything and more people than ever would buy my products.

Looking at Table 2 and the figures (which were excellent) confirms that that is what happened. The medians did not change much at all, but some of the top quartiles went down. In addition, the p values in this table seem to be off. Some tiny differences were highly sig. (e.g. sugars in desserts).

But I propose a fundamentally different analysis: I think quantile regression should be used, with time as the main iV and good group as a covariate. For the longitudinal analysis, a multilevel quantile model could be used. 

Peter Flom

Reviewer #3: Changes on the nutrient content of packaged foods and beverages after the initial implementation of the Chilean Law of Food Labelling and Advertising: a nonexperimental prospective study.

The impact of regulation of foods is currently of great public health interest and this piece of work examining the introduction of mandatory labelling on processed foods and drinks is therefore welcome. However, I have some concerns over the analysis and manuscript. The chosen analyses methods could be stronger as the results presented show simply pre- and post-law differences that fail to consider existing trends in nutrient content. It is possible that these differences would have been seen if the law had not come into effect not least because the Chilean government signalled its intent to use regulation through the modification of their tax on SSBs. This should be clearly stated in the limitations section. 

Title. 

Change "on" to "in" i.e. "Changes in the nutrient content of packaged foods and beverages after the initial implementation of the Chilean Law of Food Labelling and Advertising: a nonexperimental prospective study".

Abstract.

Background

The first sentence of the background section doesn't make sense; suggest changing to something like: "In June 2016, the Chilean Food Labelling and Advertising Law that mandated front-of-package warning labels and marketing restrictions for unhealthy foods/beverages was implemented. 

Methods

Sometimes the time periods are referred to as T0 sometimes as T0 please be consistent.

Although later specified it is not immediately clear what T0: Jan-Feb 2015/2016 refers to. Suggest clarifying this earlier. Perhaps reword the first sentence in the methods to state that data collection was carried out at two time periods. 

N= is written twice - remove one.

There is a semicolon before N= in T1 but not in T0, change to be consistent.

First sentence of the methods should include a hyphen after pre.

The first sentence is too long. Suggest starting a new sentence after T1. As it stands it doesn't make sense particularly. Change to something like "A longitudinal subsample was also studied to explore changes in the median content of energy and nutrients of concern (total sugars, saturated fats and sodium, per 100g/100mL) and in the proportion of products with energy and nutrients exceeding the cutoffs of the Law (i.e., 'high in' products).

Findings

"High in" has not been defined, suggest changing to "product high in unhealthy nutrients".

"Significant decrease" should be changed to "Significant decreases".

Change the latter part of the penultimate sentence to "whereas changes in the amount of energy and saturated fats were uncommon." writing "content of energy" suggests a binary state of energy or no energy - which is unlikely to occur.

Interpretation.

Remove the final "s" from "sugars".

Author Affiliations 

What are the numbers in the author affiliations? Are they postcodes / zip codes? 

Line 21 why is "Stefanie Vandevijvere Senior Public Health Nutrition Scientist" written?

The author affiliation for institute 1 does not match that of the corresponding author. 

Introduction

Line 53 "In the last decades is too ambiguous, if this manuscript is being read in 20 years time it will not make sense. Please be more precise e.g. "From 1990-2020, packaged foods..."

Personally I find the use of ultra processed foods unhelpful. That is, the implicit suggestion that simply processing the food makes it worse than not processing the food. The law referred to in this manuscript addresses nutrients rather than the number of processes that are required to produce the final food. The second sentence states that it is the high content of energy and nutrients linked to NCDs that is the issue. I am aware of the two positions of investigating diet either by examining total foods or looking at nutrient composition. But the ultra-processing part feels too poorly defined. Sure sausages and processed meat are linked to increased risk of NCDs but then this law focuses on the nutrient composition. Line 59 further discusses improving nutritional quality by reducing nutrients of concern rather making foods less processed. In short could you not write ultra-processed foods in the first sentence of the introduction as you appear to be interested in nutrients and should therefore write that the package foods and beverages contain more nutrients of concern rather than are "processed". 

Line 60 change content to level or amount. Again content is like contains - does it contain nutrients of concern? Yes - it has energy.

Line 61. I think we have been interested in the nutritional quality of packaged foods for longer than "since the 2000s". 

Line 65 change contents to content.

Line 65 change i.e. to e.g. as these are examples of possible strategies rather than the exhaustive list of all possible strategies implied with i.e.. Also include "adding" (or something similar) before "upper limits for sodium content" as it can be read as banning trans fats or banning upper limits for sodium content.

Line 67 "However, as most of these strategies are voluntary the impact on the nutritional quality of the overall food supply has been limited". I disagree. There are lots of mandatory strategies that incentivise reformulation such as mandatory back of pack labelling, for example, fortification of wheat with calcium, folate and iodine fortification, SSB and fat taxes and VAT/GST. Similarly there are restrictions on more classical food safety grounds like the lead and arsenic content of foods for example. Recommend that this sentence is amended to make the point I think you are trying to make which is that without mandating these strategies industry is less likely to do what is wanted and therefore government intervention is required. 

Line 70 feels a bit cumbersome to refer to the law by its full title every time. Could you use an abbreviation?

Line 73 I suspect the implementation was not agreed in a staggered way rather the law was to be implemented in a staggered way, please amend accordingly. 

Line 75 sentence starting "Such products..." which products? This seems out of context or at least needs changing to fit.

Line 76 Not sure what the "Chilean experience" is. Should this be something like "The Chilean government has introduced a wide range of strategies to improve public health and tackle childhood obesity that combines a FOP label in combination with child obesity prevention strategies..."

Line 76 child obesity should not be hyphenated. 

Line 79 sentence starting "Therefore, in..." this could do with tightening up. Suggest something like "Therefore, in the current study we aimed to study changes in the proportion of 'high in' products and changes in energy and nutrients of concern in packaged foods and beverages, before and after the initial implementation (i.e., <1 year of the first phase cutoffs) of the Chilean Law.

Line 81 1 year does not need hyphenation.

Methods

Line 87 pre does need a hyphen

Line 87 as above T0 and T1 have a comma before N=... Please be consistent.

Line 87 the colon after T0 is subscripted it should not be.

Line 89 was the Law only implemented in Santiago or are you only looking at Santiago? Clarify the interest in Santiago ideally give context of Santiago with the rest of Chile. 

Line 89 new sentence after Chile.

Line 90 hyphen after pre 

Line 94 are you interested in ingredients? Is the amount of each ingredient given on the label otherwise you won't be able to see reductions in the amount other than perhaps a change in the order of the ingredient (any way I suspect without amounts/percentages it's going to be inexact). 

Line 94 why were you taking photographs of products before the law was introduced? Was this work for something else or were you aware that the law was going to be introduced? If it is the former then it's not a big deal and perhaps just include a sentence addressing this if it's the latter, that is, the authors (and manufacturers) were aware in 2015 the law was going to be introduced then your baseline sample could be biased.

Line 96 This had to be read a couple of times to get the meaning perhaps "(one supermarket from each of the 6 major chains in Chile)"

Line 96 could you give an indication of the market share of these 6 chains for those unfamiliar with Chile? 

Line 96 are the 3 candy distributors localised to the high income areas? Could you add more clarity to this as I can only think of shops that distribute to a local area and that any distributor would be at a wider geographic area.

Line 99 could you define packaged foods? Is that essentially all foods? Does it include packaged fruit and vegetables?

Line 99 did you explore alternatives to visiting stores such as manufacturer or supermarket websites? The wayback machine can be useful for retrospective website access.

Line 105 did you carry out any validation of the data, for example we saw a temporal changes in product labelling when manufacturers went from providing nutritional information per 100g to providing nutritional information as made up with semi-skimmed milk. This resulted in dramatic reduction in sugar and increases in protein for the same unreformulated product. Possible checks could be to examine the atwater factors (e.g. fat = 9kcal per 1g) and see if based on the macronutrient content energy is as expected. 

Line 116 i.e. should be changed to e.g. Also the list provided (nuggets and others) does not provide a lot of clarity, simply writing non-sausage meat products seems like a rather odd classification where I am unsure what is special about sausages but at least seems easy to understand whereas the examples of nuggets and others make it more not less ambiguous. 

Line 121 I am unclear why you included two years of data in the baseline sample. The issue with this is - and I'll come to this in the analysis section - is that when you carry out a simple pre post study like this you don't know why the changes occured. They may be due to the law but they may simply reflect on-going reformulation in the food market and that these differences would have appeared if the law had not come into effect. Comparing data from 2015 to 2017 means that the changes are more likely to have come about due to the general reformulation towards healthy products. In short could you explain the benefit for including two years of baseline data as it is not immediately apparent. 

If the product was available in 2015 and 2017 but not in 2016 then either it was not captured in 2016 or it was withdrawn and relaunched. If it is the former then it casts doubt on the completeness of the sample.

Also why not do two years post implementation i.e. 2018? Including two years pre-implementation data seems to confuse things, raise questions and add little. It feels like the 2015 data have been included simply because they were collected and are available.

Line 123 (i.e., 2016 items were included) suggested rewriting to something like "only items collected in 2016 were included" to remove ambiguity 

Line 129 As I understand it you are only interested in individual products with greater than 1% market share. What percentage of products have less than <1% market share? Having 100 products on the market means that all of these (except for a couple) would be excluded - or have I missed something?

Line 145-146 line spacing looks smaller than the remainder of the document

Line 149 food densities, more technically known as specific gravity.

Line 152 Would help if the two analyses that are presented (i.e. cross-sectional and longitudinal) had their own subsections 

Line 161 law should be lower case.

Line 163 hypen for pre

Line 166 "distributions were non-parametric". This is incorrect and should be changed. I suspect what is meant here is that the distributions were not normally distributed and therefore non-parametric tests were used. Distributions have parameters, tests that do not rely on these parameters are non-parametric. 

Line 167 Was there any adjustment for multiple testing? If one component of the food changes then the others are more likely to do so. Similarly manufacturers may make the decision to reformulate across their entire product range and therefore these tests are not independent and this should be addressed accordingly - typically be lowering the alpha threshold.

Line 167 Analyses should not have a capital letter as the second word in the subtitles does not anywhere else.

Line 174 insert "at" between available and each.

Line 174 remove space after /

Line 175 see earlier point. For food groups with 482 products the market share for the majority will be <1% whereas the group with only 69 will have a greater number of products with 1+% market share (probably) therefore the results will be biased towards categories with fewer groups. 

Line 176 should it be "in" before "the standard..."?

Line 180 should be analyses

Line 184182 error

Results

Is it possible to have a summary of all changes carried out? For example if 5 groups have shown a decline but 5 have increased then there may be no overall benefit. Or perhaps not all food groups are equal as small changes in foods that are highly consumed may have a greater impact on health than large changes in foods that are rarely consumed. Similarly is there a hierarchy in the changes that were observed that is: are changes in energy more important than changes in sodium for example?

Line 187 would prefer % after each percentage value e.g. 51% to 44%.

Line 188 "left columns" is unclear please use column numbers or change.

Line 189 can you provide confidence intervals?

Line 206 change left column to figures on the left or something similar.

Line 209 see above. It is unlikely that foods will be reformulated for a single nutrient. If reformulation occurs it is likely to change energy and nutrients.

Line 214 should be "Changes in".

Line 214 It is unclear what is meant by this sentence, changes...were observed for products in which there was no...change.

Line 219 please write Figures in full each time.

Line 220 remove "overall".

Line 228 change "on" and "among" to "in" and as appropriate elsewhere.

Line 229 change "among" to "in" and as appropriate elsewhere.

Line 223 change to (by between 1% and 6%, depending on the food group)

Line 214237 error.

Discussion

Line 243 change to "This is the first study to evaluate changes..."

Line 247 what is meant by "relative" decrease?

Line 250 this is written to appear that the groups were the driver of the change whereas the groups are passive and had change done to them. 

Line 255 "significant improvements were seen".

Line 257remove has been described (or change to have) 

Line 259 change "supply" to "suppliers" if appropriate. 

Line 259 change "...menus of fast..." to "menus in fast..." 

Line 261 remove "the distribution of".

Line 262 remove comma after USA.

Line 262 change first "on" to "of" second "on" to "in".

Line 263 references after the end of the sentence.

Line 268 "given a previous report..." this sentence is not clear please rewrite.

Line 271 add "the" before "healthy school food environment".

Line 272 agree that changes can be brought about with incentives however this cannot be continued indefinitely. 

Line 277 change "in their way to implement" to "on their way to implementing".

Line 280 remove "between".

Line 281 change "2015 Chilean packaged food supply in Chile" to "packaged food supply in Chile in 2015" or something similar.

Line 284 change "have" to "has".

References

Some of the journals are abbreviated and some are not. For example Obesity Reviews is given in full and as Obes Rev

American journal of public health requires capital letters for all words.

Some article titles have capital letters for each word some do not.

Page numbers are not included for all articles e.g. Ref 2 includes page numbers Ref 6 does not.

Page number for Ref 1 should read 10-19 not 10-9

Refs 7, 30, 32 only have 1 page number (2/836/65)

Table 1

Remove extra line space in Savory baked products - High in Sodium and Sausages - High in Sodium.

Table 2

Remove space between 100 and mL

Footnote "n" is lower case - change to upper case for consistency.

Why are there sometimes [ and sometimes (?

Sometimes there is a space and a long dash sometimes no space and hyphen. Please be consistent.

Please use consistent number of decimal places.

Figure 1 

Not sure why the title is all in capitals, 

Subsample and exclusions are spelled incorrectly. 

Sometimes the thousands digits are seperated by a comma sometimes they are not - please be consistent. 

Some of the lines do not join into the middle of the boxes, sometimes there are spaces between the boxes and the lines. 

Recommend that the final boxes are aligned. 

Sometimes there are spaces after the = sign, sometimes there are not - please be consistent. 

The first horizontal line looks slightly skewed, that is not perfectly horizontal as it has a step in it. 

Sometimes the description is given before the N sometimes after - please be consistent. 

Change the boxes with N T0=10,081 and N T1=8,563 to match the others by changing the position of the N i.e. T0: N=10,081 ...

In the final boxes T0 and T1 are not sub(sub)scripted

Figure 2

Please include all plots (Energy, Sugar, Sat Fat, Sodium) for each category

n= has become lower case here, please use the same case throughout.

Use T0 and T1 rather than Pre and Post

Include a footnote to state why the numbers (n) aren't the same for energy and sugars 

Why do the density curves begin at different points <0?

Remove space between 100 and mL for consistency.

The resolution of these plots is not great - can you change to vectorised images?

The contrast between the green and blue is not great - consider more contrasting colours though maybe this would be improved with resolution changes.

Consider shading the area under the plot

Why is there white space above some of the curves in some plots and not others? Ideally the same scale would be used but failing that at least make them look similar.

Same with the x axis - some curves fill the x axis whereas others have lots of space.

Breakfast cereals - it appears that some breakfast cereals have (close to) 0kcal of energy. Is this correct?

Should be non-sausage meat products throughout the manuscript.

Use "Cheese" or "Cheeses" throughout the manuscript for consistency.

Where are the plots for Ready-to-eat meals?

Supplementary Table 1

Please make line spacing the same for both solids and liquids.

Why are square brackets used here for the first time - please be consistent.

No spaces after 100

Supplementary Table 2

The foods seem to be ordered by random - could you make it alphabetical or are they by number of products?

These categories need lots of revision.

Unclear what the difference between "flavored water with sugar" and "aromatized/flavored waters with sugar" is.

Are "fantasy beverages" a Chilean product?

The "other" group should go at the end of the list to catch those products not categorised previously.

There are three categories for "fantasy beverages" with sugar, without sugar and other. The first two categories cover all possible options - what goes in the final category?

What is the difference between "100% fruit juices" and "100% fruit juices (no added ingredients"? Surely if it is 100% fruit juice then it is only fruit juice and has nothing else. 

Flavoured water with sugar is repeated but spelled differently.

What are milk drinks? That seems to encompass all of the subsequent categories.

Change "skim" to "skimmed" for consistency.

Yogurt is in the Milk and milk-based drinks category and in the yogurt category.

Remove spaces around /.

Light or diet yogurt includes light or diet yogurt with fruits and/or nuts.

Breakfast cereals - change to "dry fruit cereal bars".

The three cereals categories seem very specific i.e. balls, honey stars and flakes. Are there a lot of honey stars cereals in Chile?

Dessert & ice cream - change to vegetable chips (remove s)

Candies & sweet confectionery - why are nuts in here?

Remove s from popcorns

Confectionery is spelled two ways - confectionery and confectionary.

Why are chocolate chips in sweet spreads not "Candies and sweet confectionery"?

What is "glazed"?

Jam (all kinds) includes the other jam category.

What is "other products used for pastry"?

Savory baked products - two spaces after "sopaipillas,"

What is the difference between "packaged white bread" and "packaged white bread loaf"?

What is the difference between " packaged whole wheat bread loaf (regular, light or diet)" and "packaged whole wheat bread loaf (all kinds)"?

"Corn torillas" is repeated.

Should be "vegetable broth" not "vegetables broth".

"Other dressings" and "other dressings for salads" do not appear to be mutually exclusive - please amend.

Savory spreads - remove spaces after / to be consistent with elsewhere.

What is the difference between "fresh cheeses and light cheeses" and "fresh cheeses"?

Why is bacon and ham in the sausage category?

Sweet baked products: are bizcochos and biscochos different things or is that a typo?

"Frozen breaded meat" and "frozen marinated or seasoned meat" are written twice

What is the difference between "nuggets (chicken or turkey)" and "nuggets".

Unclear what the difference between "Soups for one or other type of instant soups" and "instant soups" is. 

Supplementary Table 3

The first sentence isn't clear, "outlier" should be changed to "outliers" and the final part " considered unlikely to happen for the specific group were omitted" needs rewriting. 

Would be helpful to state that these values are per 100g or serving or whatever the units are

N is lower case unlike in the majority of the document - please be consistent

ii should be "Sugar" not "sugars".

iii should be "fat" not "fats".

[LINK]

---

## [Decision Letter · Decision Letter 2]

5 Jun 2020

Dear Dr. Corvalan,

Thank you very much for re-submitting your manuscript "Changes in the amount of nutrient of packaged foods and beverages after the initial

implementation of the Chilean Law of Food Labelling and Advertising: a non-experimental

prospective study." (PMEDICINE-D-20-00319R2) for review by PLOS Medicine.

I have discussed the paper with my colleagues and the academic editor and it was also seen again by three reviewers. I am pleased to say that provided the remaining editorial and production issues are dealt with we are planning to accept the paper for publication in the journal.

[LINK]

We look forward to receiving the revised manuscript by Jun 10 2020 11:59PM. 

Sincerely,

Adya Misra, PhD

Senior Editor 

PLOS Medicine

plosmedicine.org

Requests from Editors:

Comments from AE: 

I think to understand the impact of a labeling process, one needs to look carefully at the specific distributions and where the line is set for something to be declared "high". In the case of sugars in milk drinks, for example, the distribution is fairly flat and the line for being declared high is in the middle of the distribution. There is a clear shift in the distribution post-introduction of labeling. Other distributions are fundamentally different and the line of where "high" is are sometimes way above the top end of the distribution pre-labeling. It is no wonder that there is little impact. In addition to making the changes that have been proposed in this paper, I would rather hope that any resubmission might highlight more clearly some things like the above, which are messages that don't seem to be coming through very strongly at the moment.

Please remove the original submission file from the submission system as can erroneously be used to generate the full submission PDF by the system

In the abstract- its not clear what you mean by a longitudinal subsample that was also analysed. Please clarify and revise as needed

Line 114- perhaps you can revise to 20th century using roman numerals?

Line 143- I think you mean 30% of the population in Chile lives. Please revise

Did your study have a prospective protocol or analysis plan? Please state this (either way) early in the Methods section.

Please remove page numbers and line numbers from STROBE checklists provided as these are likely to change. Please use paragraphs/sections instead. 

The financial information can be removed from the word document as it is automatically pulled from the article meta-data. Please only provide the information in the financial disclosure section

Same goes for the data availability information, which can be provided in the data statement

In the discussion you say that your results suggest that food and beverage reformulation occurred rapidly. I’m not sure if this can be directly linked, as you have noted in the limitations section, so I recommend removing this sentence and carefully toning down the discussion to avoid overstating your conclusions

Please add a sentence in the Competing Interests statement to acknowledge that Barry Popkin is an Academic Editor for PLOS Medicine.

Comments from Reviewers:

Reviewer #1: The authors have addressed my previous comments. The quantile analysis has strengthened the paper and made it easier for readers to understand the changes occurred around the cut-off. Only a few minor comments left:

Line 114 what is XX? 

Line 167 "small packages where not collected" It is unclear what it is meant. 

Line 257 How are "food/beverage groups" controlled/ assessed in the quantile regression? 

Reviewer #2: The authors have addressed my concerns and I now recommend publication

Peter Flom

Reviewer #3: The manuscript is much improved. I have a few, minor comments. 

Line 76 - school sales at school. Seems like the first "school" is not required

Line 114 - add in years instead XX

Line 165 - no comma after i.e.

Line 166 - add for after asked or change asked to chosen

Line 167 where should be were

Line 173 no comma after i.e.

Line 274 two spaces after all

Line 273 remove hyphen from sweet baked

Line 337 associated with a decrease not associated to 

Line 380 reports of improvement not to improvement

Line 403 change in to on.

Line 412 add reference for the National Dietary Survey

Line 405 put such in front of as

Lines 410-415 417-424 combine in to single paragraph

Line 446 remove the 

Line 642/670/736 Cutoff corresponds - add s

Line 643/671/737 comma after i.e.

Line 681/747 McNemar tests

Table 1 - What do the confidence intervals represent? Should this be interpreted as, for example, 26% of beverages were "high in" any category in T0? If so this doesn't require a confidence interval as it is a single value with no variance. Also some of the point estimates do not lie exactly between the confidence intervals, e.g. sweet baked products - high in sugars 95 (90-97). 

Table 2

Savoury snacks sodium T1 - remove bracket

Sometimes one or two decimal places are given sometimes there are none.

Desserts & ice creams sugars - for p25 and p50 the values at T0 and T1 are the same and the p-value is 1 whereas for p75 T0 and T1 are the same but the p-value is 0.30. Is this because of rounding?

Table 3 

Beverages Any 'High in' the dash is different to the others

Ready-to-eat meals, please check n as elsewhere it is 109

Table 4 

Candies & sweet confectionary is used, whereas confectioneries is used elsewhere.

Savory spread sugars p25 T0 = 0, T1 = 0 but p!= 1 please add decimal places to explain.

Fig 1 

Include a line linking total products collected to products collected in 2015

Some of the arrows are not exactly horizontal

Left hand side exclusion box

Change n to lower case

change semi-colon to comma for consistency

Right hand side exclusion box - remove colon and change semi-colon to comma

Fig 2 

Unclear why the x axis start below 0. Are there negative numbers given or is this the default plot? 

Kcal does not begin with a capital K elsewhere in the document 

In some instances the n provided in the plot does not match that given in table S2, for example Fig 2C n for yogurt is 181 whereas Table S2 states 184. Similarly n for cheeses = 60 and 61 in S2, Fig 2M T0 n=242 Table S2 states 243. This may be due to some products not providing all information if this is the case then perhaps provide a footnote.

S2 Table.

Beverages T0 n is 686 whereas Table S2 states 688

Flavoured is spelled both with and without a "u" suggest the American English spelling without a "u" to be consistent.

Milks & milk-based drinks - too many uses of the word "powder". 

Yogurts - no n after Longitudinal.

Breakfast cereals - flakes not lakes.

Desserts & ice cream - vegetable not vegetables and chips not chip.

Candies & sweet confectioneries - change carcass for coating.

Sweet spreads - Unclear on the difference between "honey" and honey (all kinds).

Savory baked products - suggest starting all sections with a capital letter but if not Christmas should have one. Is Christmas bread savory?

Savory spreads - should hicken be chicken? Unclear why broth is in the spread category not soups.

Ready to eat meals - Longitudinal should be on new line

 S3 Table 

There is a semi-colon in the first list (after ice creams) that makes it unclear which groups you are referring to. Perhaps a value is missing? Also a full stop after sausages that should not be there. Oxford comma after confectionery in iv. No space after > before 8000mg

[LINK]

---

## [Editor Report · Decision Letter 3]

24 Jun 2020

Dear Dr. Corvalan, 

On behalf of my colleagues and the academic editor, Dr. Nicholas J Wareham, I am delighted to inform you that your manuscript entitled "Changes in the amount of nutrient of packaged foods and beverages after the initial implementation of the Chilean Law of Food Labelling and Advertising: a non-experimental prospective study." (PMEDICINE-D-20-00319R3) has been accepted for publication in PLOS Medicine. 

PRODUCTION PROCESS

PRESS

PROFILE INFORMATION

Thank you again for submitting the manuscript to PLOS Medicine. We look forward to publishing it. 

Best wishes, 

Adya Misra, PhD

Senior Editor 

PLOS Medicine

plosmedicine.org